# Preconditioning of sediment failure by astronomically paced weak-layer deposition

Xingxing Wang[1,2], Vittorio Maselli[3], Luca Flessati[4], Hongbin Wang[5], Zhilei Sun[5], Qing Wang[1], Jie Chen[6], Qing Li[5], Stefano Alberti [7], Markus Kienast [8], Shucheng Xie [9] & Qiliang Sun [1,2,9] ✉

Low-strength sediment layers within continental slope strata precondition submarine sediment for failure, potentially leading to destructive tsunamis. Using geophysical and Ocean Drilling Program well data, here we show that the glide planes of widespread submarine failures in the northern South China Sea, dated to the glacial stages following the Mid-Pleistocene Transition, have higher opal content, particle size, and porosity, which reduce the undrained shear strength. Cyclic weak-layer deposition, modulated at Milankovitch time scale, was controlled by increased ocean primary productivity and sedimentation rates linked to high-amplitude sea-level fluctuations and intensified winter monsoons. This study represents an important step forward for understanding how climate influences the formation of weak layers and the stability of continental slope globally.

Submarine landslides and associated mass-transport deposits (MTDs) are common features along continental margins. Their surficial extent and sediment volume span several orders of magnitude, reaching sizes that are much larger than the onshore counterparts[1]. The sudden movement of submarine landslides can also generate catastrophic tsunamis, potentially impacting coastal communities and infrastructures, with enormous environmental and economic consequences[2,3]. Submarine landslides normally detach from one or multiple basal surfaces, often associated with "weak layers". These weak layers are characterized by remarkably low shear strength, driven by changes in one or more sediment properties, such as lithology, particle size, porosity, permeability, fossil type/abundance[4–7]. A weak layer can extend laterally for thousands of square kilometers and may represent the basal surface for multiple landslides over a wide area, as demonstrated along the eastern flank of the Eivissa Channel in the Mediterranean Sea[8,9]. In the past, it has been proposed that the formation of weak layers, and thus the occurrence of submarine landslides, may be driven by the impact of climate change on continental weathering, sea-level fluctuations, and ocean dynamics[7,10]. However, a mechanistic understanding of the processes linking climate change to slope instability along continental margins has yet to be established.

Here, we use seismic reflection data tied to Ocean Drilling Program (ODP) wells to show how the occurrence of widespread submarine landslides in the Quaternary sediments of the northern South China Sea (SCS) can be linked to the climatic and eustatic variations that the region experienced after the Mid-Pleistocene Transition (MPT), a global climate event that occurred between ~1.2 and 0.7 Ma[11,12]. We demonstrate that continental weathering, monsoon intensity, and oceanographic processes modulated at ~100-kyr-long glacial-interglacial cycles[13–18] resulted in increased supply of coarse-grained sediments to the continental slope and production/preservation of biogenic silica, which led to the formation of weak layers and thus preconditioning sediments for failure.

[1]Hubei Key Laboratory of Marine Geological Resources, China University of Geosciences, Wuhan, China. [2]Laboratory for Marine Mineral Resources, Qingdao Marine Science and Technology Center, Qingdao, China. [3]Department of Chemical and Geological Sciences, University of Modena and Reggio Emilia, Modena, Italy. [4]Faculty of Civil Engineering and Geosciences, Delft University of Technology, Delft, The Netherlands. [5]Key Laboratory of Gas Hydrate, Ministry of Natural Resources, Qingdao Institute of Marine Geology, Qingdao, China. [6]School of Geosciences, China University of Petroleum (East China), Qingdao, China. [7]College of Forestry, Oregon State University, Corvallis, OR, USA. [8]Department of Oceanography, Dalhousie University, Halifax, Nova Scotia, Canada. [9]State Key Laboratory of Geobiogeology and Environmental Changes, China University of Geosciences, Wuhan, China. ✉e-mail: sunqiliang@cug.edu.cn

## Results

### Regional setting

The study area is located along the continental slope of the northern SCS, offshore the Dongsha Island, at water depths ranging from 1500 to 4000 meters (Fig. 1). The SCS, the largest marginal sea situated at the convergence of the Eurasian, Pacific and Australian-Indian plates (Fig. 1a), experienced rifting, drifting, and subduction throughout the Cenozoic[19]. The tectonic stresses during the rifting and seafloor-spreading stage (~65–16 Ma) resulted in rugged topography visible on the modern seafloor with numerous northeast-to-southwest-oriented ridges[20]. In the northern SCS, the peak of tectonic activity occurred approximately between 10.5 and 3.6 Ma, driven by the subduction of the SCS plate beneath the Philippine Sea plate, which focused tectonic stresses predominantly on the continental shelf and upper slope[21]. This

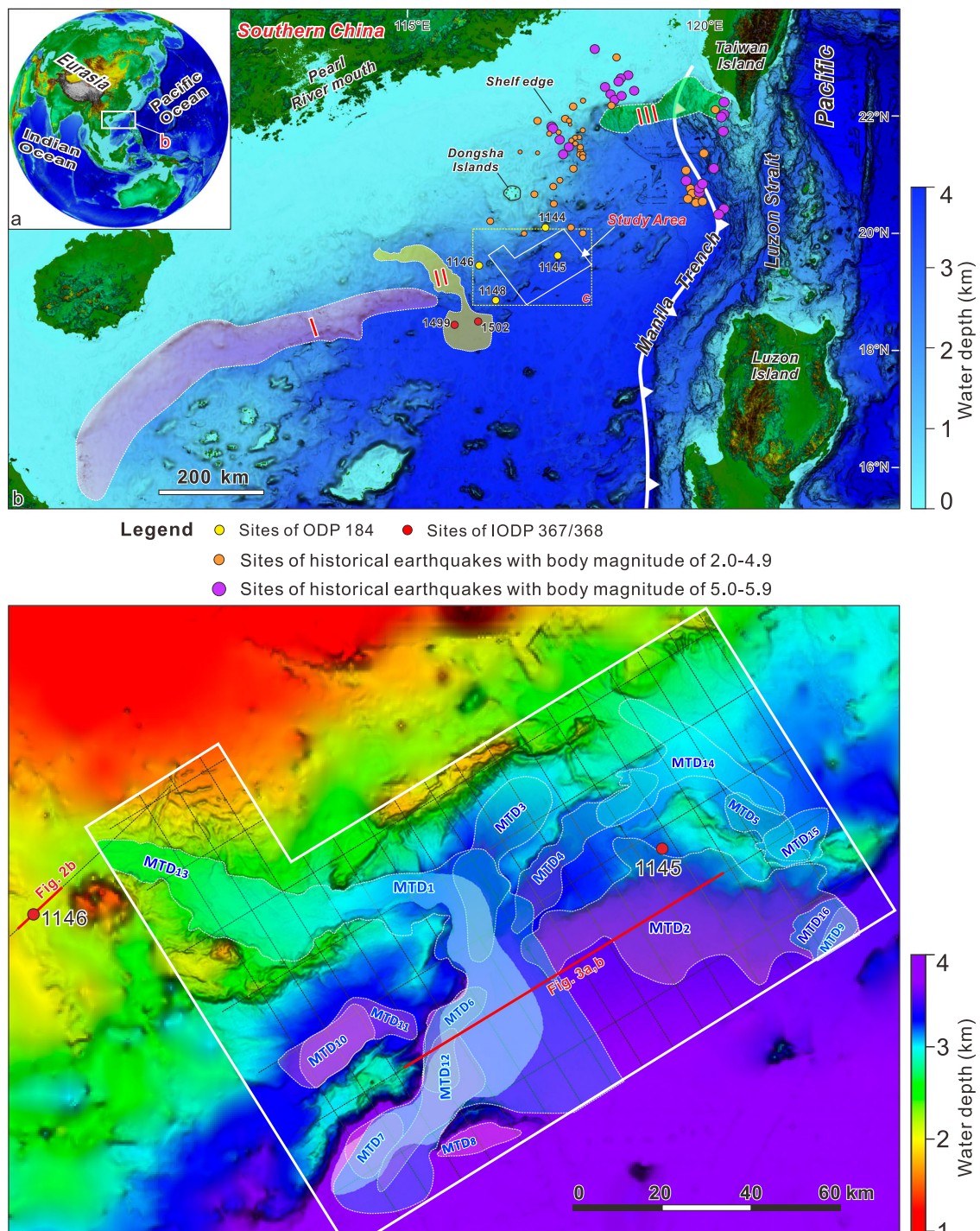

**Fig. 1 | Geological background and distribution of the mass-transport deposits (MTDs) in this work. a** Geography and tectonic elements of the South China Sea (SCS), **b** Regional map of the northern SCS. The colored areas I-III represent the documented landslide-dominated regions in the northern SCS. **c** Bathymetry map of the study area with 16 MTDs distribution highlighted by white dashed lines. The seismic profiles are shown as black dashed lines with the ones presented in Figs. 2 and 3 as red lines. The historical earthquakes in (**b**) are from the ref. 59. The white solid boundary in c outlines the high-resolution bathymetry map of the study area.

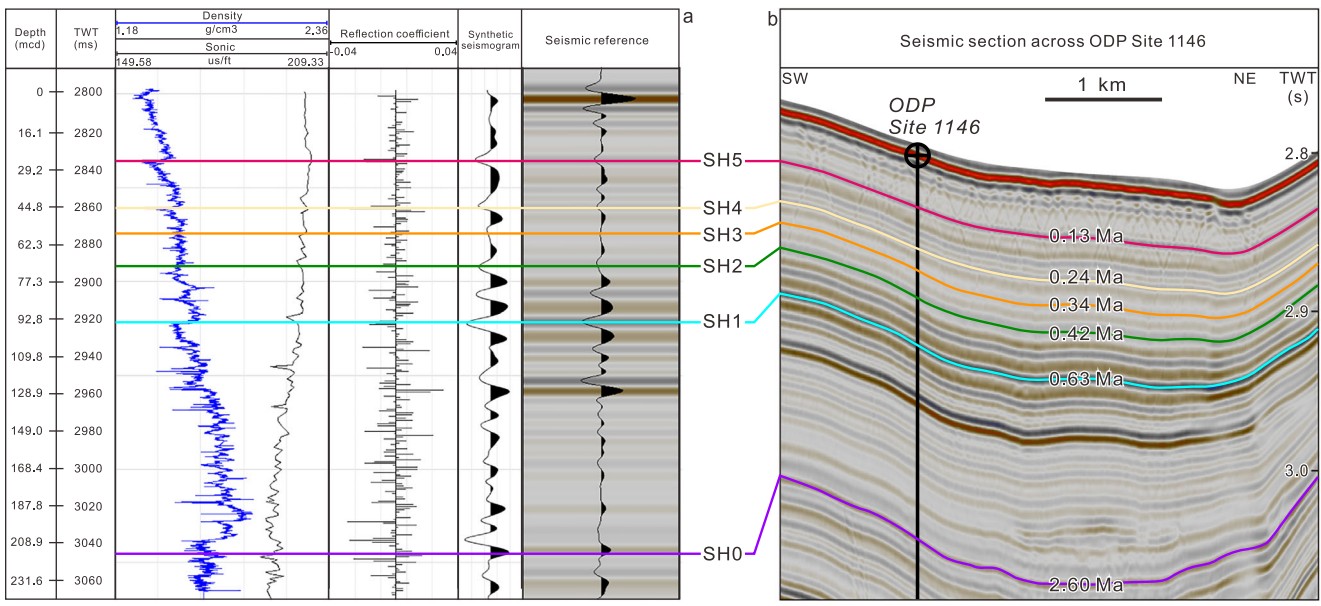

**Fig. 2 | Dating of the critical seismic horizons based on the seismic-well tie at ODP Site 1146. a** Seismic-well tie for the ODP Site 1146, **b** Seismic profile across the ODP Site 1146 with the highlighted seismic horizons SH0 to SH5 and their ages.

regime greatly weakened at the beginning of the Quaternary, and since then, the region has experienced a general tectonic quiescence[22].

During the last 1 Myr, terrigenous sediments deposited on the northeastern margin of the SCS were mainly sourced from Taiwan and Luzon islands through the Deep Water Current circulation (in water depths >1500 m), and from southern China via riverine supply (e.g. the Pearl River) as well as through eolian inputs driven by the East Asian winter monsoon[23,24].

Submarine landslides and MTDs, particularly involving Quaternary sediments, have been identified along the northern margin of the SCS (Fig. 1), indicating generalized slope instability phenomena[25,26].

### Chronology of key seismic horizons

We identified six regionally extensive seismic horizons, named SH0 to SH5 from bottom to top, which are tied to ODP Site 1146 (Fig. 2). The deepest seismic horizon, SH0, is associated with a prominent high-amplitude positive-polarity reflection that ties with ODP Site 1146 at ~216 meter composited depth (mcd). Micro and nano fossil assemblages were used to date ODP Site 1146[27], and SH0 marks the base of the Quaternary, encompassing sediments yielding an age of ~2.6 Ma.

Horizons SH1 to SH5 are marked by a high- to medium-amplitude and negative-polarity reflection, corresponding to conformable surfaces at the location of ODP Site 1146. SH1 at a depth of ~96 mcd has an age of ~0.63 Ma (marine isotopic stage 16, i.e. MIS 16) and approximates the end of the MPT[28]; SH2 at a depth of ~71 mcd has an age of ~0.42 Ma (MIS 12); SH3 at a depth of ~59 mcd has an age of ~0.34 Ma (MIS 10); SH4 at a depth of ~46 mcd has an age of ~0.24 Ma (MIS 8); and, lastly, SH5 at a depth of ~26 mcd has an age of ~0.13 Ma, corresponding to MIS 6 (Fig. 2). Outside of ODP Site 1146 and towards deeper waters, horizons SH1 to SH5 often show an erosional character, particularly when they correspond to the basal glide planes of submarine landslides, as discussed in the following sections (Fig. 3, Figs. S2–S7 in Supplementary Information).

### Seismic stratigraphy of the northern SCS and distribution of MTDs

By using well-established criteria for the analysis of seismic reflection data[29–31], we identified four seismic facies (SF) that are diagnostic of the main depositional environments in the study area: hemipelagite

deposits (SF1), sediment waves/contourite deposits (SF2), coarse-grained turbidites/submarine fans (SF3), and MTDs (SF4) (see the details in the Section 2 of the Supplementary Information).

Based on seismic facies SF4, we identified sixteen MTDs in the post-SH1 stratigraphic succession and only one MTD in pre-SH1 deposits (Figs. 1, 3). These MTDs have variable spatial extent, with the smallest being of ~73 km² and the largest greater than 2500 km². The maximum thickness ranges from 32 ms two-way-time (TWT) (~25 m) to 305 ms TWT (~235 m) (Figs. S2–S7 and Table S1 in Supplementary Information). Horizon SH1 corresponds to the basal surface of six MTDs (named MTD 1 to MTD 6), of which MTD1 and MTD2 have a surface area greater than 1000 km². Horizons SH2 and SH3 are covered by MTD12 and MTD13, the latter accumulating over a ~2850-km²-wide area and reaching a maximum thickness of 145 ms TWT (~112 m). Three MTDs formed at different locations above horizons SH4 and SH5 (Fig. 3, Figs. S2–S7 and Table S1 in Supplementary Information). Overall, the 16 MTDs have an areal extent of ~6163 km² (the overlapping area is counted only once), presenting 46% of the whole study area (Fig. 1c). These MTDs are mainly characterized by sharp transitions with the adjacent un-deformed strata (e.g., Fig. 3, Figs. S3 and S4 in Supplementary Information), similar to the frontally confined MTDs described in ref. 32.

### Lithological properties of ODP Site 1146

ODP Site 1146 was used to quantify how sediment properties vary with depth, particularly across the horizons SH1-SH5 (Fig. 4). In the interval 131–96 mcd, the porosity generally increases upward from 55% to 70%, accompanied by a saturated bulk density decrease from 1.7 g/cm³ to 1.5 g/cm³ (Fig. 4b). In parallel, the fraction of coarser-grained end member, defined as the grain size mode of 19 μm following the end-member modeling in ref. 33, displays a remarkable increase during the glacial periods above 101 mcd. SH1, corresponding to the lowstand of MIS 16 (Figs. 4, 5), marks the first large-amplitude increase in opal content, grain size and porosity. SH1 is also overlain by low-porosity and clay-rich sediments (Fig. 4), which can explain its negative polarity signature in seismic profiles (Fig. 3). The same sedimentological changes are observed at horizons SH2 to SH5, at depths of ~71 mcd, ~59 mcd, ~46 mcd, and ~26 mcd, respectively (Fig. 4). The ratio of smectite/(chlorite+illite), an important indicator for the East Asian monsoon[33], displays an upward increase trend from ~160 mcd to

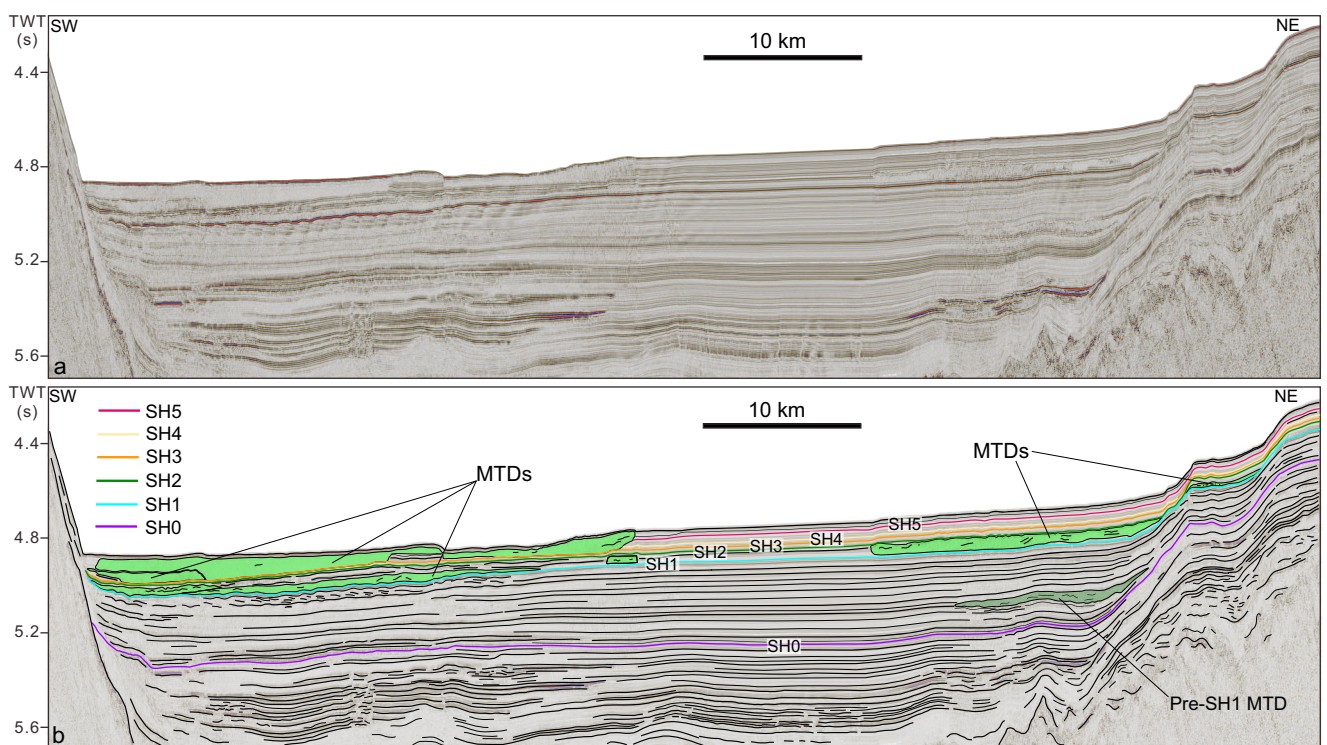

**Fig. 3 | The seismic profile showing the critical seismic horizons and MTDs in the study area. a** Un-interpreted seismic profile, **b** interpreted seismic profile (location in Fig. 1c).

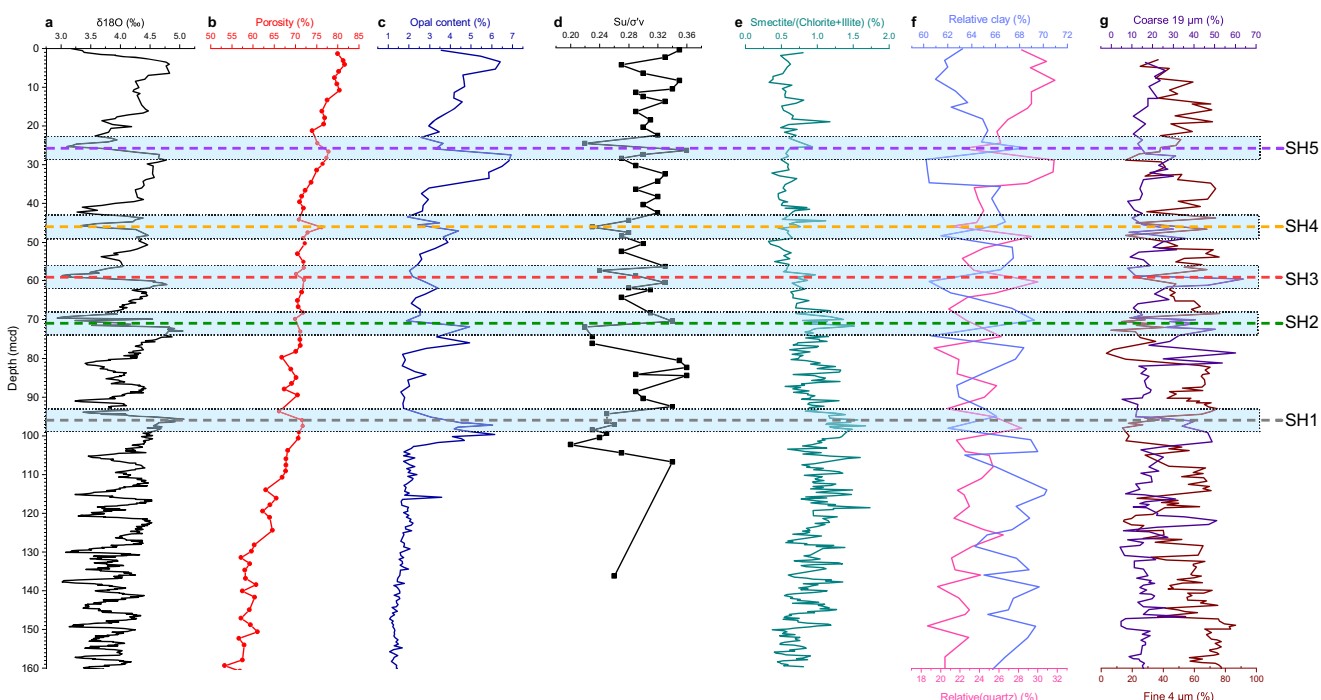

**Fig. 4 | Depth variations of sedimentary properties at ODP Site 1146. a** $\delta^{18}O$[67], **b** porosity[27], **c** opal content[35], **d** the ratio of $S_u/\sigma'_v$, **e** smectite/(chlorite+illite) ratio[68], **f** relative quartz and relative clay data[66], **g** the data of coarse 19 μm and fine 4 μm data[33]. The blue bars with dashed outlines indicate the depth uncertainties for the horizon SH1–SH5.

~96 mcd, and so before MIS 16, and an upward decrease from ~96 mcd to the seafloor (Figs. 4, 5).

The stratigraphic interval deposited after MIS 16 (above SH1) is characterized by cyclic variations in lithological parameters, such as opal content, sedimentation rates, particle size, and clay content as indicated by the percentages of coarse- and fine-grained end members (with a grain size mode of 4 μm; Fig. 4). As indicated by the $\delta^{18}O$ values (Figs. 4, 5), these cyclic variations follow the glacial-interglacial cycles. Furthermore, the changes in $\delta^{18}O$ become more prominent after MIS 16, indicating that the northern SCS experienced higher amplitude sea

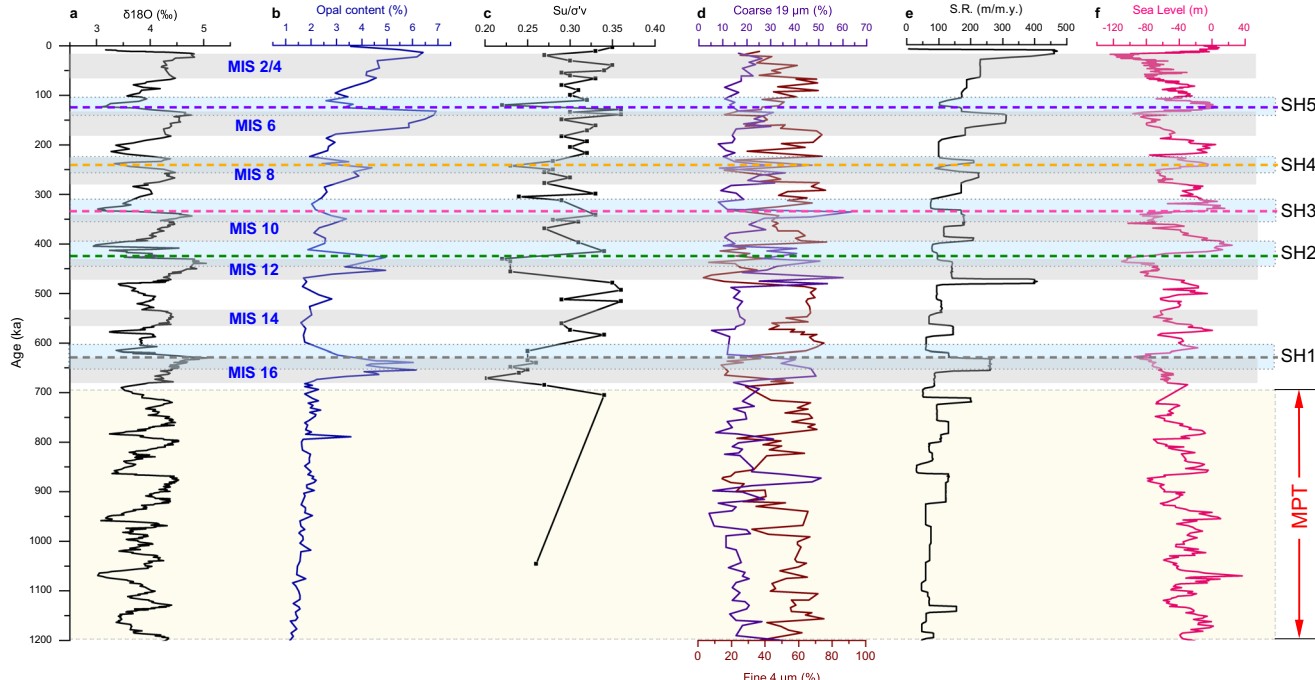

**Fig. 5 | Time variations of sedimentary properties and deposition background at ODP Site 1146. a** $\delta^{18}O$[67], **b** opal content[35], **c** the ratio of $S_u/\sigma'_v$, **d** coarse 19 μm and fine 4 μm data[33], **e** sedimentation rate, **f** sea level[69]. The average sedimentation rate is calculated based on the time variation of $\delta^{18}O$. The gray bars without outlines indicate the marine isotopic stages; the blue bars with doted outlines indicate the time uncertainties for the horizon SH1-SH5. The bars with dashed outlines indicate the MPT (i.e., middle Pleistocene Transition) interval.

level oscillations after the MPT (Fig. 5), as also seen globally in other oceanic basins[11]. Glacial periods are overall dominated by increased amounts of opal, reaching about 7% in weight of total sediments, larger sediment grain size, and lower clay content (Fig. 5). The opal is primarily composed of diatoms and radiolarians (>90%; See Fig. S8 in Supplementary Information), with a smaller contribution from sponges[34]. Glacial intervals are also characterized by higher sediment accumulation rates, reaching up to 460 m/m.y. (Fig. 5e). Similar variations of these sedimentary properties are also observed at ODP sites 1144, 1145, and 1148 in the northern SCS[33,35].

## Geotechnical analysis

Geotechnical analyses on sediments from ODP Site 1146 were conducted to estimate the depth variation of the sediment shear strength and, consequently, to detect weak layers.

It has been widely observed that the shear strength of a soil containing diatoms is affected by the diatom content. Notably, when the diatom content exceeds 25% by mass, the interlocking of diatom particles contributes to an increase in shear strength[36–42]. However, for lower diatom contents, generally less than 10–25%, the geotechnical behavior of soil-diatom mixtures in terms of shear strength, compressibility, and index properties is practically unaffected by the presence of diatoms[37,38,42,43]. In fact, when a small quantity of diatoms is mixed into the soil matrix, the diatom particles do not interact with each other[37]. This prevents the interlocking and, thus, any increase in shear strength.

In this study, the maximum observed opal content is approximately 7% (Fig. 4b), with only about half attributed to diatoms[34]. This indicates that diatoms do not interact sufficiently to enhance shear strength and, consequently, we can reliably estimate the shear strength of the in situ sediments by using standard soil mechanics methods. To characterize the mechanical behavior of the sediments, we analyzed 59 samples of ODP Site 1146 from the Kochi Core Center in Japan, collected at various depths ranging from 0.4 to 137 mcd within

and outside the presumed weak layers. Direct shear strength estimations obtained using, for example, triaxial tests, cannot be performed on such samples as they are disturbed; the cores are approximately 26 yr-old and likely lost the original water content. Therefore, we adopted a well-known approach introduced by Skempton (1957) for normally consolidated clays, which suggests that the undrained shear strength ($S_u$) could be preliminarily estimated from the plasticity index ($PI$, defined as the difference of the liquid limit and plastic limit) and the in situ vertical effective stress ($\sigma'_v$)[44]. See details in Fig. S10 in the Supplementary Information, and Tables in the Supplementary Data 1 and 2. According to this approach:

$$\frac{S_u}{\sigma'_v} = 0.11 + 0.0037PI \tag{1}$$

The variation of the non-dimensional ratio $S_u/\sigma'_v$ with depth is plotted in Fig. 4d. This ratio is particularly useful for identifying variations in soil strength attributable to changes in soil composition. The experimental data clearly show local minima of $S_u/\sigma'_v$ at the depths of SH1-SH5 (horizontal lines), suggesting the presence of weak layers at those depths (Figs. 4d, 5c).

## Principal component analysis and interpretation

To unravel the dominant control of the sediment variation from high-dimensional data over the past 0.7 Myr, we performed a principal component analysis (PCA) for the nine sedimentary elements (i.e., relative contents of quartz and clay, opal content, smectite/(chlorite +illite), contents of coarse end member (with grain-size mode at 19 μm) and fine end member (with grain-size mode at 4 μm), sedimentation rate, porosity and $\delta^{18}O$ as a proxy for sea level) derived from ODP Site 1146.

The first two principal components (PC1 and PC2) explain 48.84% and 20.80% of the original parameters, respectively (Fig. 6). PC1 is positively related to $\delta^{18}O$, quartz, opal, sedimentation rate, and

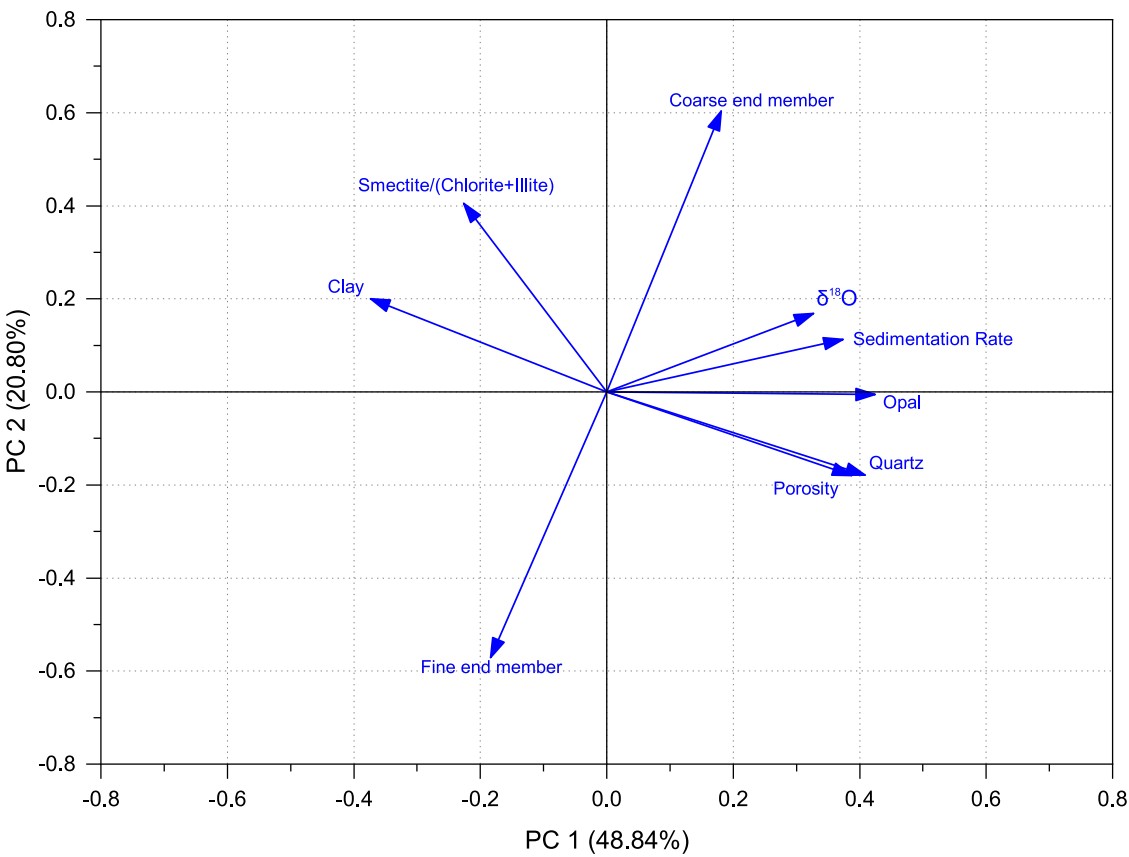

**Fig. 6 | Principal component analysis loading plot of PC1 and PC2 that explains the controls of sediment variation in the past 0.7 Myr.** The X- and Y-axis represent the element loading of PC1 and PC2, respectively. See the detailed element loadings in the Table S2 in Section 4 of the Supplementary Information.

porosity but negatively influenced by clay (Fig. 6). It displays good correlation to glacial-interglacial cycles, with larger values during sea level lowstands and enhanced variation amplitude above the horizon SH1 (so after the MPT), indicating PC1 is greatly influenced by sea level fluctuations (Fig. 7). PC2 is positively related to the coarse end member, smectite/(chlorite+illite) and negatively related to fine end member (Fig. 6, Table S2 in the Supplementary Information). The ratio of smectite/(chlorite+illite) generally increased with the intensification of East Asian summer monsoon (EASM) on the northern SCS margin[45,46]. The close relationship between PC2 and smectite/(chlorite+illite) indicates that PC2 could reflect EASM intensity. The upward decreasing trend of PC2 suggests that the EASM weakened overall since MIS 16 (Fig. 7). The increase in the coarse end member could be attributed to two possible factors: 1) enhanced amplitude of sea level falling during lowstand intervals, or 2) intensification of EASM[33]. The higher values of the PC2 correlate with the greater content of coarse end member during glacial periods (Figs. 5, 7), when the climate were characterized by weaker EASM and stronger East Asian winter monsoon (EAWM)[14]. This suggests that PC2 is also influenced by sea level fluctuations. In essence, our analysis indicates that, after the MPT in the past -0.7 Myr, the delivery of coarse end member towards the deep SCS was primarily driven by sea level oscillations under the background of a weakening EASM.

## Discussion

Globally, the MPT ( -1.2–0.7 Ma) marked a fundamental reorganization of the Earth system that led to longer glacial-interglacial climate cycles (from 41 kyr to 100 kyr), larger global ice volumes, and higher-amplitude oscillations in temperature, atmospheric $CO_2$, and sea level[11,17]. The increased amplitude of sea level oscillations, up to 120 meters, promoted the exposure of broader continental shelves[47],

which resulted in higher sediment production and delivery into deep-water, particularly during glacial lowstand intervals (see discussions in references[48,49]). Similar changes in sea level and sediment supply after the MPT have been also observed in the northern SCS[50,51], corroborating the PCA results (Figs. 6, 7).

Data from ODP Site 1146 indicate that post-MPT clay-rich sediments primarily accumulated during sea level highstands (indicated by low $\delta^{18}O$ values: Fig. 4), likely as a consequence of reduced supply of land-derived material to the continental slope due to the greater distance between river mouths and the shelf edge[52]. Conversely, the accumulation of coarser sediment fractions, whose production can be intensified by a stronger winter monsoon[14,24], increased during glacial sea level lowstands. During these periods, the shelf was subaerially exposed, allowing rivers to directly feed sediments into the slope[53,54].

Glacial intervals also experienced increased availability of nutrients, promoting an increase in marine primary production[55,56]. An intensified glacial winter monsoon also favored the intrusion of nutrient-rich Northern Pacific currents into the northern SCS[35]. This increased nutrient availability likely contributed to an increase in the production of siliceous microfossils (e.g., radiolarians, diatoms, and sponge spicules), as recorded in ODP Site 1146 and other sites in the region[27,34] by higher opal content (Fig. 5). Moreover, biogenic silica burial efficiency and associated opal preservation are generally higher when sedimentation rates are higher[57]. Consequently, the significant increase in glacial sedimentation rates (Fig. 5e) likely contributed to higher opal concentrations during glacial periods in the northern SCS following the MPT.

A study focusing on submarine landslides primarily from the North Atlantic and the Mediterranean Sea found no statistically significant correlation between landslide frequency and changes in sea level or sediment accumulation rates[58]. Although this study did not

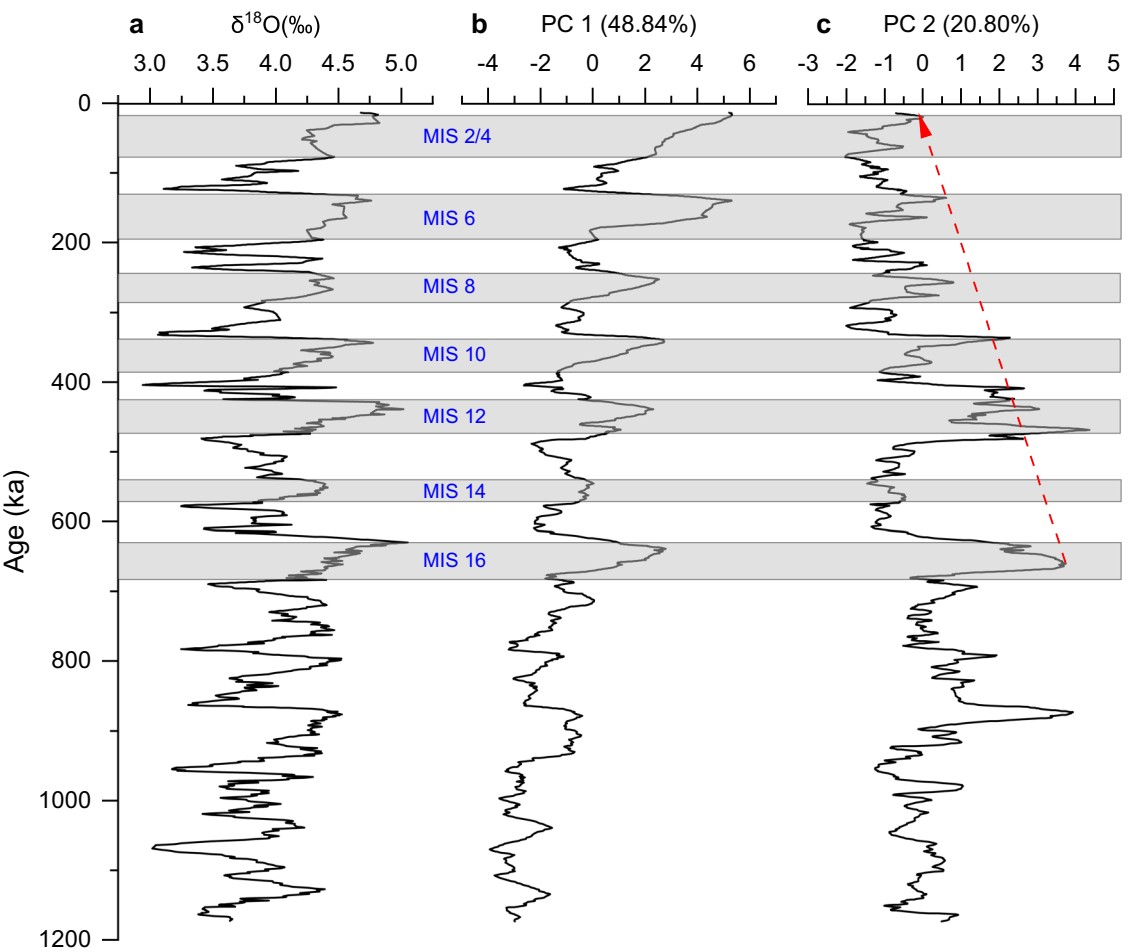

**Fig. 7 | Variation over time of PC1 and PC2. a** δ[18]O, the agency of ages and glacial-interglacial cycles, **b** PC1, **c** PC2. The shaded bands indicate the glacial intervals.

include examples from the northern SCS, it is plausible that similar conclusions apply to this region. Therefore, factors other than sea level fluctuations and sediment supply must play a critical role in driving the observed post-MPT increase in landslide occurrence.

One such factor is the presence of biogenic opal within sediments, which has been shown to reduce shear strength by increasing sediment porosity when the opal content is lower than 10–25%[37]. In the northern SCS, opal-enriched sediments accumulated cyclically during glacial intervals following the MPT, forming stratigraphic layers with lower shear strength than the overlying and underlying units deposited during sea-level rise and highstand intervals (Figs. 4d, 5c). These opal-enriched layers tie in ODP Site 1146 with regionally extensive seismic horizons (SH1–SH5), which mark the basal surfaces of a series of landslide deposits (Figs. 3, 5). Although the northern SCS margin is currently tectonically quiescence, low magnitude earthquakes[59], mostly occurring more than 200 km northeast of the landslide area (Fig. 1b) may induce sediment shaking. Taken together, these lines of evidence suggest that the opal-enriched layers likely acted as mechanically weak intervals that preconditioned sediment for failure under the influence of external triggers such as low-magnitude earthquakes or increased sediment supply[60,61]. This mechanism also explains why MTDs in the northern SCS are primarily, if not uniquely, observed in the late Quaternary sediments deposited after the MPT (Figs. 3, 8).

Submarine landslides detaching from a basal, weak layer deposited under the influence of specific climate conditions, such as those observed in this study, have been identified elsewhere. For example, the submarine landslides offshore northwest Africa moved along

surfaces made of diatom-rich layers accumulated after the MPT[5,62]. The cyclic formation of weak layers related to post-MPT climate change has also been observed in carbonate settings: In the Comoros archipelago (western Indian Ocean), low-density layers with high calcite content and prone to liquefaction have been found along the slope of Mayotte, deposited during glacial intervals since MIS 12[10]. Three Neogene submarine landslides have been recently discovered along the continental slope of the eastern Ross Sea in Antarctica, overlying diatom oozes and glaciomarine diamicts, which have deposited in response to climate-driven variations in biological productivity, ice proximity, and ocean circulation[63]. These examples further illustrate the role that climate change played in the formation of weak layers and the occurrence of deep-water mega-slides, also in near-seafloor sediments.

This study identifies sixteen submarine landslides within Quaternary sediments of the northeastern margin of the South China Sea. The base of these landslides can be traced along five distinct glide planes, corresponding to as many seismic horizons, which correlate in ODP Site 1146 with the glacial intervals that occurred after the Mid-Pleistocene Transition. The oldest of these horizons formed during MIS 16, the first high-amplitude drop of sea level in the Quaternary. At ODP Site 1146, each horizon corresponds to a sedimentary layer characterized by increased opal content, grain size, and thus porosity, which we demonstrate contributed to the reduction of undrained shear strength and, consequently, to the formation of weak layers. The observed lithological variations were mainly driven by enhanced sea level fluctuations and intensified winter monsoon on Milankovitch time scales during the glacial intervals following the MPT. Our study provides strong evidence for a link between submarine landslides and

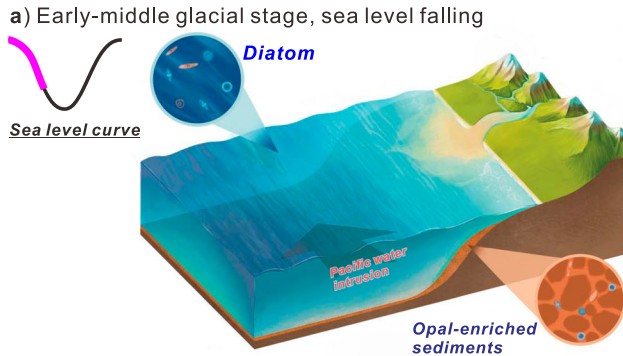

**a**) Early-middle glacial stage, sea level falling

*Sea level curve*

*Diatom*

*Pacific water intrusion*

*Opal-enriched sediments*

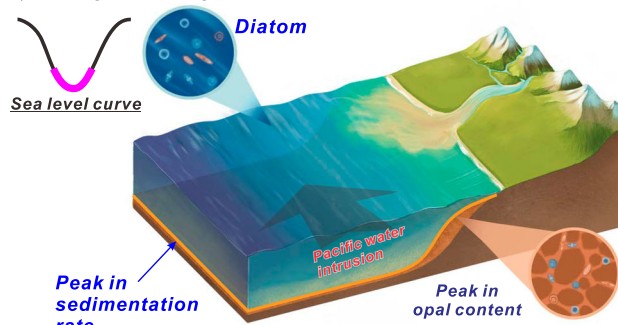

**b**) Peak glacial stage, sea level lowstand

*Sea level curve*

*Diatom*

*Pacific water intrusion*

*Peak in sedimentation rate*

*Peak in opal content*

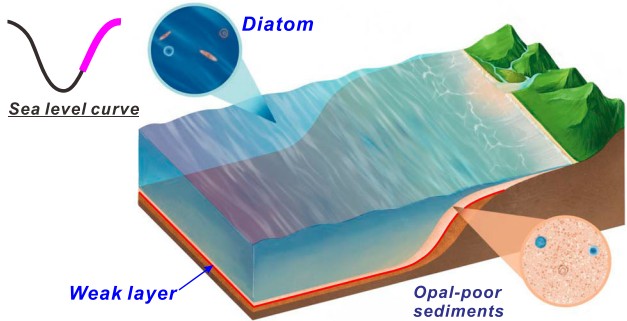

**c**) Interglacial stage, sea level rising to highstand

*Sea level curve*

*Diatom*

*Weak layer*

*Opal-poor sediments*

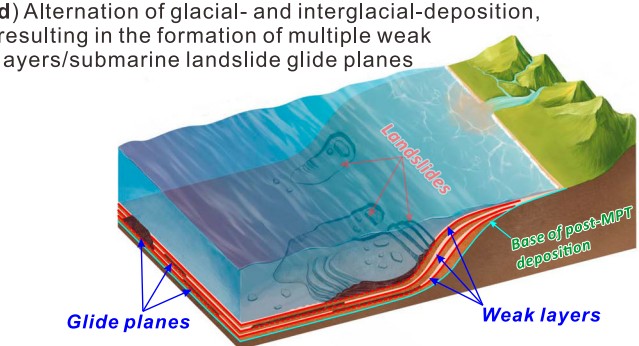

**d**) Alternation of glacial- and interglacial-deposition, resulting in the formation of multiple weak layers/submarine landslide glide planes

*Landslides*

*Base of post-MPT deposition*

*Glide planes*

*Weak layers*

**Fig. 8 | Conceptual model for the deposition of weak layers.** Opal-enriched layers accumulate during falling sea levels (**a**) until glacial lowstands (**b**), when the peak in opal content and sediment supply is reached. Opal-poor layers accumulate during sea level rising to highstands with low sediment supply (**c**). The weak layers are shown at the interfaces between glacial and interglacial sediment layers (**c**, **d**). The size of arrows in (**a**) and (**b**) indicate the intensity of Pacific water intrusion into the SCS.

climate change, with implications for assessing the location and scales of potential submarine landslides in deep-water regions across the global ocean. The successful use of the plasticity index for having a preliminary estimation of the undrained shear strength suggests that existing/disturbed sediment samples, such as ODP/IODP, can be effectively utilized to advance this research.

## Methods

### Bathymetric and seismic data interpretation

Geophysical data used in this study were newly collected and processed by the China Geological Survey. The high-resolution multibeam bathymetric data, which cover an area of ~6600 km² and have a grid cell resolution of 200 m, were acquired using a SeaBat8150 multibeam system, with a frequency of 12 kHz and maximum coverage angle of 100°. The data were used to map the distribution of submarine landslides with a morphological expression at the seafloor (Fig. 1c). Also, the gridded bathymetry data set (i.e., GEBCO_2024 Grid) from General Bathymetric Chart of the Oceans is used to show the geographic background of the northern SCS margin[64] (Fig. 1a, b).

Two-dimensional (2-D) multichannel seismic reflection profiles, which have a total length of 2236 km, were acquired using air guns with a total capacity of 550 cubic inches and a 3-km-long streamer with 240 channels. Shooting and sampling intervals are of 12.5 m and 4 ms, respectively. The seismic profiles are arranged with a grid of 8 × 16 km (Fig. 1c). The dominant frequency of the seismic data decreases with depth, and the frequency for the near-seafloor sections could reach up to 77 Hz. The laboratory-measured velocity for the shallow strata (0–200 meters beneath seafloor, mbsf) of the northern SCS margin is about 1500–1600 m/s[27,65], allowing the corresponding seismic resolution to be ~5 m. The seismic profiles, displayed in SEG polarity, with seafloor reflection indicated by positive polarity, were used to identify

the distribution and characteristics of MTDs, as well as the six regionally extensive seismic horizons (i.e., SH0-SH5) across the study area (Fig. S2–S7 in the Supplementary Information). This study primarily maps the MTDs that can be identified on at least two intersecting seismic profiles and have surficial extents greater than 10 km². Seismic interpretation for different MTDs was all conducted using the software *Petrel 2015*.

### Seismic-well tie and dating of seismic horizons

ODP Site 1146 is located along a seismic line that connects to the seismic lines in the study area (Fig. 1c). ODP Site 1146 displays continuous Quaternary sedimentation with a high sedimentation rate, and thus records high-resolution variation in sediment properties with climate changes[27]. This site has abundant and completed data on the well logs (e.g., *Natural Gamma Ray, Density*), P-wave velocity measurements, lithology (e.g., quartz, feldspar, clay mineral, carbonate, and mean grain size), oxygen isotope, porosity, opal content and organic carbon (Fig. 5)[27,66–68]. Based on the density and P-wave velocity, we made a synthetic seismogram of ODP Site 1146 and made a seismic-well tie at this site. In this way, the depths of the critical horizons could be obtained in this study (Fig. 2). Combining with the oxygen isotope data-based age-depth model[67], we dated the ages of the critical seismic horizons.

ODP Site 1145, at water depth of 3175.5 m, is located ~1.5 km away from a seismic line used in the study area (Fig. 1c). Given that the Quaternary strata is not disturbed by landslides at ODP Site 1145 (Fig. S9 in Supplementary Information), this well also could be used to investigate the physical and lithological properties of Quaternary sediments. Although the P-wave velocity below 58 mcd is slightly overestimated due to an operational error, we obtained the inverted P-wave velocity for this site based on the simplified three-phase

equations (see the details in Supplementary Information). The inverted P-wave velocity and laboratory-measured density are used to make the synthetic seismogram of the ODP Site 1145 and conducted a seismic-well tie. In this way, the depths of the critical horizons at ODP Site 1145 could be obtained in this study (Fig. S9 in Supplementary Information).

Through correlating the high-resolution δ18O records of the planktonic foraminifera (*G. ruber*) in ODP Site 1146[67] and the LR04 stack[28], we could define the ages of seismic horizons at ODP Site 1145. The dating results of critical seismic horizons at the two sites (i.e., ODP Sites 1145 and 1146) are consistent. Considering ODP Site 1145 has a very low sedimentation rate at certain intervals (e.g., 2 cm/kyr for the interval of 86.8–88 mcd), our analyses for sediment properties are mainly based on those from ODP Site 1146, supported by the sediment properties from ODP Site 1145.

### Atterberg limits

59 sediment samples of ODP Site 1146 have been obtained from the Kochi Core Center, Japan, between 0.4 and 137 mcd (see Tables in Supplementary Data 1, 2 and Fig. S10 in the Supplementary Information). Each sample consists of the sediments extracted from a continuous 10-cm-long section of the core. In the geotechnical plots (i.e., Fig. 4d, Fig. S10 in the Supplementary Information), the depth of each sample is the depth in the middle of the 10 cm-long section. These samples were used to measure Atterberg limits (i.e., liquid and plastic limits), and the measurement have been performed at the Institute of Rock and Soil Mechanics, Chinese Academy of Sciences.

The liquid and plastic limit were measured using the cone penetration method, following the standard for geotechnical testing method (GB/T 50123-2019). Each sample was evenly mixed with distilled water by slow stirring, and a standard cone (80 g, 30° angle) was lowered into the soil surface, allowing it to sink for 5 s. The depth of penetration was then measured. Water was either added or removed (by drying the sample) to achieve a penetration depth of 15–17 mm. At this point, the water content was determined by weighing the sample before and after drying it at 60 °C. This procedure was repeated to measure the water content corresponding to penetration depths of 11–13 mm, 7–9 mm, and 3–6 mm, respectively.

For each sample, the four penetration depths and their corresponding water contents were used to perform linear fitting on a log-log scale, with water content on the horizontal axis and penetration depth on the vertical axis. The fitting quality was evaluated using $R^2$ values, and the result was accepted when $R^2$ was equal to or greater than 99%. Based on the linear fitting line, the liquid limit and plastic limit were determined as the water content values corresponding to penetration depth of 17 mm and 2 mm, respectively. The plastic index then was derived from the difference between the liquid and plastic limit. The liquid limits, plastic limits and plastic indexes are shown in the Supplementary Data 1.

## Data availability

The seismic and high-resolution bathymetry data utilized in this study were provided by the China Geological Survey. The dataset are protected and are not available due to data privacy laws. The GEBCO bathymetry data are available at https://www.gebco.net/ data-products/gridded-bathymetry-data. The Drilling data from ODP Site 1145 and 1146 are available at https://web.iodp.tamu.edu/OVERVIEW/. The Atterberg limits generated in this study are provided in the Supplementary Data 1 and 2.

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

## Acknowledgments

We are grateful to the China Geological Survey for giving access to the seismic and multibeam bathymetry data and allowing the publication of this work. We sincerely thank Dr. Delia Oppo for providing $\delta^{18}O$ data of ODP Site 1145, Dr. Shimin Wan for providing lithological data of ODP Site 1146, Dr. Rujian Wang for providing the data of opal content, and Kochi Core Center for providing samples of ODP Site 1146. We are grateful for the assistance of Dr. Pan Chen, Ms. Zhuoya Xie and Mr. Weihang Jiang during the Atterberg limits test. This work was financially supported by the National Natural Science Foundation of China (No. 42222607, Q. Sun; No. 42476252, X. Wang), Laoshan Laboratory (No.LSKJ202203501, X. Wang), the Natural Science Foundation of Shandong Province (No. ZR2020QD034, X. Wang), the Marine Geology Survey Program of China Geological Survey (No. DD20221707, Z. Sun).

## Author contributions

X.W. proposed the initial conceptual idea, interpreted the seismic profiles, conducted the Atterberg limit tests, wrote much part of the manuscript and outlined the figures. V.M. and Q.S. improved the conceptual idea, helped defining the relationship between climate changes and slope instability with the support of M.K. and S.X. and contributed to the writing of the manuscript. L.F. developed the geotechnical analysis and helped writing the manuscript supported by S. A., H. W., and Z.S. helped in the interpretation of the multibeam bathymetry and Ocean Drilling Program data, and analyzed the relationship between climate change and formation of siliceous fossils. Q.W. was responsible for the Principal Component Analysis. J.C. was responsible for the P-wave velocity inversion and seismic-well tie. Q.L. supported seismic data interpretation and figure drawing. All the authors contributed to the final editing and revision of the manuscript.

## Competing interests

The authors declare no competing interests.
