## [Transparent Peer Review file · Nature Communications]

Preconditioning of sediment failure by astronomically paced weak-layer deposition

Corresponding Author: Professor Qiliang Sun

Version 0:

Reviewer comments:

Reviewer #1

(Remarks to the Author)

This study first identified a dozen MTDs in the seismic record that shared one common sliding surface (H1). Then the authors attributed the climate characteristics such as greater swings of glacial-deglacial cycle, sea level etc. during the MPT as the cause of this common weak layer (H1). I have a major question on this interpretation:

On the one hand the authors argued that because of the bigger swing of sea level during the MTP there were much more terrestrial sediment discharged into the study area and therefore produced the strata of coarser sediment and larger porosity, one the other hand the author also showed that the sedimentation rate during this same period was not higher! This seems a huge self-contradiction.

The authors also said that the higher content of diatom and silicate could be used to explain the H1 layer's high porosity (and therefore weak), but did not elaborate. I recommend that authors dig a little deeper into why the break-up of diatom would increase the water content and porosity. It would be a much better contribution to our understanding the mechanisms of the common weaker layer, especially if it can be a bit quantified.

Reviewer #2

(Remarks to the Author)

October 20, 2022

This study aims to demonstrate that there is a widespread weak layer that facilitated several large submarine landslides to occur in the northern South China Sea and that this layer was deposited by changes resulting from the Mid-Pleistocene Transition.

The concept/argument that a weak layer can be formed and would be related to climate-induced changes, is sound, and if definitively shown, would likely be significant and noteworthy. However, the primary areas of concern that I find after reviewing, are (1) there are not enough sufficient details in support of the claim regarding the weak layer. Abstract line 27-28 argue for a reduction in shear strength but it is not demonstrated to actually be weak with shear strength data, modeling, or other attempts at demonstrating this. There is circumstantial evidence regarding the potential role(s) of grain size, porosity, diatom presence, that the authors point to in support of defending a weak layer, but it is circumstantial evidence. There could be sufficient data for the authors to use in support of the claims, but in its present form, the paper needs additional details to support the claims about the layer being weak. (2) the explanation of the role of H1 in the failure and the riming of failure, is very unclear.

The rationale that the layer must be weak because it underlies some slope failure deposits cannot be taken as fact and should be challenged for at least two reasons: (1) If the layer is actually a weak layer, how do the authors envision its role in the slope failures given that it the horizon (H1) is apparently conformable and mappable across all the lines? Why is it not eroded in the areas beneath the landslides and especially in the scarp areas where the slope failures occurred? The authors

map it as a continuous surface from the unfailed area at ODP Site 1145 to beneath the MTD-1 landslide shown in Figure 2. Why would the weak layer be present beneath the landslides? The weak layer itself would be mobilized and eroded once the landslide occurs. I see in the figures and supplementary figures, that it does appear in some small areas that H1 is indeed missing and eroded, but it does not appear that this is common, but rather H1 is preserved. (2) When a submarine landslides occurs, the mass moves out of the source area and eventually the mass is deposited (MTD) on the seafloor some distance from the source area, unless they are localized slumps that do not move very far from the source area. It seems the data support that former, that these are slides that mobilized away from where they originated. If true, the weak layer that the mass failed along is not the one the deposit is resting on, but rather the scarp area where the landslide originated. Do you have the data to show this, that the detachment in a scar area is H1?

If that's the case, then it begs another questions...because the seismic data show that the moving mass came to rest on H1, it is puzzling to understand what it suggests about the timing of the failure. If H1 was the seafloor at the time of failure, what is the mass itself composed of? I believe the authors envision a scenario in which H1 was deposited during the MPT, and then some time later slope failures occurred and detached along H1, which would have been now at some depth below the seafloor. It's not clear to me how this is explained, but it really needs to be examined and presented for a more convincing analysis and understanding of these landslides and the role of H1.

Another possibility that is not addressed is perhaps the layer itself is not weak, but somehow causes a weakening of the strength characteristics of the layers immediately above it? These specifics of what the data show and what is being interpreted really need to be addressed more robustly. There is no description of any shear strength data at Site 1145 or for Leg 184. Based on my review of the ODP database, there unfortunately appears to be no shear strength measurements acquired. This is unfortunate as shear strength is usually a standard measurement collected. The authors need to present this information and try to work around this particular challenge, if you want to stick with the claim that it is in fact a weak layer. Ideas that come to mind would be to request core samples across the H1 surface. The cores are presumably too old to offer much hope for direct shear strength measurements, but you would do other laboratory studies to address the shear strength characteristics of that horizon. If you want to argue that diatoms are breaking and causing pore water pressures to spike (and therefore weakening the layer), you could examine the diatoms under microscope to address that possibility. There could also be a modeling effort to assess slope stability of these materials and these slopes. There is also insufficient evidence about the line-to-line correlations that the H1 horizon is indeed the same across all the lines. How were these done?

Other items of note:

- Where are the sonic and bulk density logs coming from?
- Why do you need to model the P-wave velocity and elastic parameters when you have the sonic and bulk density?
- What wave are you convolving with the reflection coefficient series?
- It is hard to understand the first paragraph in the Results Section. Particularly in Lines 86-89, it seems incomplete: "For the other five MTDs that did not shared the basal surface of H1, four of them (MTD8, 10, 11 and 12) superimposed on the older MTDs (MTD1, MTD4 and MTD5) that had basal surface of H1 (Figs. 1C, S7 and S8)." Please explain what you mean by "superimposed on" and what is the significance/relevance of this?
- If the H1 surface can be shown to be a weak layer, what then was the triggering mechanism(s) and when did the failure(s) occur?
- It appears to me that in the seismic data of Figure 2 that H1 at Site 1145 has a lower reflection coefficient and therefore amplitude than it does as mapped beneath MTD1. It would be good to explain or present reasons for what is controlling this amplitude response.

In summary, I believe the authors do have an interesting dataset and have made some compelling observations. However, in my assessment, there needs to be more robust work to present a clear and convincing understanding of these slides. I would be pleased to review a revised version.

Sincerely,
Derek Sawyer

Reviewer #3

(Remarks to the Author)

The manuscript by Wang and co-authors presents further results on a compelling aspect on precursors on submarine mass-wasting and their relationship to climate. This is a compelling but challenging topic to investigate, although recent works have shed further insights on it (e.g. Murlaub et al, 2018). The intent is to correlate changes in Earth's climate with sediment deposition patterns and characteristics, with a focus on bioproductivity and consequent bioclastic input to submarine basins. This is a further contribution on an important open scientific question, particularly at the current research scenario where efforts are being made to understand the influence of climate change on different natural geohazards.

The main premise of the work is to provide further proof on the effect of bioclastic elements on the location of weak layers and intervals, namely after the Middle Pleistocene Transition (MPT). The results are plausible and presented through the integration of multi-scale observations from seismic and IODP data to support the premise.

Although the main idea is broadly explained in the manuscripts current form, it feels it may be further extended to provide a more valuable and impactful work. There is quite a bit of room to expand both text and figures within the journal format, and that should be explored. For example, some of the figures currently in the supplementary material could be part of the main

manuscript. A relative sea-level curve, often included on such studies, could be presented as well for a better correlation with the climatic shifts the authors focus on, and would complement the descriptions on the text.

The authors could also test if the premise of the climate-derived failure precursor represented by H1 is also valid for other common detachment levels of other MTDs, as for example H3. Although I understand that the main premise is the shift in conditions observed in the vicinity of H1, it would still be of interest to assess if posterior high productivity episodes occurred at earlier times and if they had the same influence.

I take the chance to raise a point, which may or may not be followed, regarding the detachments. The widespread use is to refer to a weak layer, but questions had been raised if using a detachment/basal interval would be more appropriate, especially since the vertical heterogeneity between alterations of fine, coarse and bioclastic material influence the failure. On the seismic resolution here presented the detachment is generally limited to a single reflection/surface, but since there is higher resolution data available from the borehole is it possible to have further insights on this?

An aspect that can be further expanded is how these changes effectively lead to higher proneness of failure. There are mentions to previous works on the aspect of shell breakdown and dewatering but seems to be limited reference to what is available for this study that could attest for it. Another discussion point that could be pursued is how the high opal levels seem to be offset from H1. Are there any implications from this in terms of failure inception. i.e., is the weak "layer" at the high diatom level or below? As there is well data available, perhaps some aspects on geotechnical properties at these specific weak layers/intervals could be expanded, if published or available?

As a final consideration, aspects of the discussion could focus on more margins affected by mass-wasting contemporaneous with the MPT. This is, after all, the premise that the reader gets from the title of the submission. Even if other examples do not attest for identical preconditioning conditions, it will be worth summarizing which other climate-driven factors can influence widespread landslides.

It is my opinion that the manuscript idea is relevant but needs further work for a wider global impact. The authors are, of course, welcome to pursue topics or aspects other than the commented above to lead to manuscript improvements.

With best regards,

Davide Gamboa

Version 1:

Reviewer comments:

Reviewer #1

(Remarks to the Author)

The authors made substantial changes in the revised ms, one of which is to recognize H1 to H5 as weak layers, in contrasting to the claim of H1 as the only and common weak layer in the first submission. Additional data were also added and more analyzing work were provided. Those additional work sufficiently answered my two questions: 1) it clarified the original concern of "large sediment supply but non-higher sedimentation rate" contradiction; and 2) the authors decided not to further elaborate the diatom break-up issue other than citing published papers, therefore it becomes a moot point.

It's worth pointing out that the issue of "whether the weak layer can be preserved in the seismic record after a landslide gliding over it" should be fully explained by the authors. That's beyond my knowledge base so I will have the other reviewers to judge whether they are satisfied with the authors' arguments in the revised ms.

Reviewer #2

(Remarks to the Author)

Attached letter

[Editorial Note: This attachment is appended to the end of the file]

Reviewer #3

(Remarks to the Author)

The revised version of the manuscript by Wand and co-authors addressed the previous comments by reviewers, and added new data and results that, in my opinion, greatly improved the scope of the link between climate-influenced sedimentation, its role in weak layer development and geohazards. I have a few additional comments on this version which mostly consist on some clarifications.

Title: In my opinion, the title lacks an element to identify the focus on marine environments.

There are some typos or words missing here and there. This can be fixed with a thorough proof read.

Line 76: maybe I'm being picky, but can the statement that "show how the occurrence of submarine landslides in the Quaternary northern SCS can be linked to the climatic variations" be miss-interpreted as the climate variation being the cause/trigger for the landslide? The sentence is valid, but can it be stated in a less ambiguous way?

Line 100: as sediment input and type is a major, or the major element of this paper, this section needs to expand on the other sediment types relevant to the work, namely the siliceous ones that tend to be biogenic. There is also a lack of information of the source of opal, as it may come from different organisms that develop in different climatic settings – this may link to the aspects that authors further highlight regarding glaciations and monsoons. H1 is the first big increase in opal, so what is its origin?

Line 148: the title and introduction point to the submarine mass-wasting as a major element of the paper, but mention to their overall characteristics ends up being limited. I am not saying that an exhaustive and boring description of them should be made, but since they are the major feature and occur due to the weak layers analyzed there should be more on the mtd relevance to show to the reader. For example, in the body of the text the link of the MTDs with the horizons is limited to “Horizons H1 to H5 represent the basal glide planes of multiple MTDs”, with then the specific number just for H1. If the cyclicity is important, its link to the several the opal-rich horizons deserves more focus. For example, the largest MTD by far is linked to H2 and H3. That is relevant, but the text only highlights H1 as significant. This type of really relevant observations could be in this point/paragraph. I think the description of the seismic facies is secondary for this level of journal and only take up space where the points above can be developed. The facies can be described in the supplementary material.

Line 190 to discussion: The specific section could start with a simple sentence with the purpose of the analysis. For example, why make the Geotechnical analysis? Picking up on your reply to Rev 2, it was a way to confirm/further support the presence of the weak layers. We only get that link at the end with the mention to H1, H2 an H5. Can it mean H3 and H4 risk being “not weak”? Does it have implications of the paper scope? H3 also seems to be at a (local?) minimum, so why not mention it?

It is not too clear what the principal components want to represent. The reader can understand the results, but clarify the main purpose to do it. In line 240, does it make more sense to mention the fine sediment, which has the lowest of all values, or the clay component?

The interpretation of the PCA (or at least part of it) and relationship with the sea level changes is repeated in the discussion, so the results section can limit itself to show and explain the data. Maybe some of the expanded interpreted PCA made in the results can be moved to the discussion, if relevant.

The discussion could start with an opening sentence to refocus the reader on the premise of the paper, ie. Temporal cyclicity, weak layers, landslides, important after MPT, then continue with the current content.

Line 314: “promoted the formation of multiple weak layers (i.e. H1-H5), which can explain why MTDs are primarily, if not uniquely, observed in late Quaternary sediments”. I don’t disagree with this, but maybe this is an additional factor and not the main one. It would be worth referring to the higher rates of landslide occurrence during sea-level regressions, if amplitude of sl changes influence this as there were changes before the MPT but apparently less significant(?) and possibly with other margin sedimentary regimes? The sed supply rate would be worth mentioning pre and post-MPT could be worth mentioning.

Line 325 to 329: at first, I wondered why suddenly carbonates and volcanoclastic material are relevant for this, but they do link to the cyclicity argument. My suggestion is to 1) keep it, adding the argument of relevance of this cyclicity for failure preconditioning in other settings, and 2) move it to after the Antarctic case. The question is, is mentioning the carbonate/volcanic examples, would it be worth also looking for other examples where cyclicity and slides have been observed, to provide a higher global impact of the results? Or keep the focus on the silica/opal examples?

With best regards,

Davide Gamboa

Version 3:

Reviewer comments:

Reviewer #2

(Remarks to the Author)

The authors have done a thorough analysis of the strength characteristics to show evidence of weak layers based on detailed laboratory work on the original material at Site 1146.

However, please fix: In Equation 1 (Line 208), the right side should be written as $0.11 + 0.0037PI$. The text has $0.11 + 0.037PI$. I checked the excel workbook in the supplemental material and it is correct there.

Reviewer #3

(Remarks to the Author)

The revised manuscript from Wang and co-authors addressed the comments by the reviewers, and present a clear and concise new version.

I have some minor comments, and you will see the majority are just some editing aspects to help make figures more supportive.

I did make a couple of comments on the discussion, as it feels to be missing just a little bit to be better integrated with all the results and great load of work that led to them.

In my opinion, it can be published after a few more very minor modifications.

See comments below.

Line 103: very minor comment, but I noticed in figure 2 and also in a figure in the supplementary material that the representation of the well top of ODP Site 1146 is some distance above the seafloor. Maybe it is my bias, but the top/cross in circle often indicates the well datum zero. In ODP/IODP, that is the seafloor, so one could expect for the well top to be coincident with the seafloor reflection – even if for a TWT section. I am aware of how things can be challenging to match synthetic, well and seismic.

If the well path in figure 2B is merely representative of location – you are not clearly showing any well tops not other clear marker, I would suggest moving the top to the seafloor. I am perfectly ok with the way the rest of the figure is presented, and the the synthetics tie in panel A.

Again, this is a minor thing, but for the sake of precise representation, I suggest doing that minor shift on the well representation. The well zero is either seafloor or drill floor (less commonly, sea level), never hanging somewhere on the water column. It may just be a matter of changing the vector edition file. Consider the same for the supplementary material figures

Line 108: again on figure 2: as the authors determined the approximate age of the reflection markers, the figure would gain more information if the age estimated for each horizon was included. This could be a simple box with the numbers inside over the corresponding coloured line. Where, up to the authors choice, but all vertically lined up close to the left side of the seismic line could be an option. Talking about the seismic line, please indicate the orientation SW NE

Add orientations to the seismic profiles of figure 3

Line 129: I suggest citing figure 1 here as well, as that is the one that shows the distribution of the 16 mtds. Figure 3 is limited to support the statement

Line 291 : again, another very small thing, but I would suggest adding a box label to the MPT marker/level in figures, or when referring to end/after, remind that it uses SH1 as a reference. Until I memorized this, I went a bit back and forth between texts and figures, and this information could be more direct to the reader.

Line 277: Urlaub et al did not find a statistical correlation... but that does not mean you cannot try to. If all the landslides relate to the weak layers, and these are influenced by climate/sea-level changes, can't that indirectly imply a relationship?

Line 290: I do not disagree with the sentence, but it seems to make it sound a bit too certain that weak beds do not exist below Late Q. I can guess that the authors may say that this is not necessarily the case, but maybe that needs to be clear. Weak layers likely exist in the sequences without the mtds, but maybe their composition prevented the high recurrence failures seen in the Late Q, and/or at the scale that can be detected on the seismic (that can open another perspective... speculate on small collapses in MPT that seismic could not pick, except for one case, but post-MPT all conditions favours bigger ones). Looking at figure 5, a very direct idea would just be the opal content. that can be approached in the discussion as a possible influence for this higher recurrence of mtds in Late Q and not before.

As a final note, I feel the discussion is falling a bit short on actually linking many of the results together, namely the link with the Geotech results. Maybe a few words on what leads opal to fail, and what favourable set of conditions occurred on this area to lead to such recurrent failures. Even if that is on the results, it would gain if being discussed. All in all, the "core" of discussion addressing the failure aspect is nearly the same length of the conclusions. To which, I may add, are good and refer to the lithological and geotechnical analysis. However, the latter are missing on the discussion.

Hope these comments are helpful, and look forward to see the work published.

With best regards,

Davide Gamboa

In this letter, we provide a point-by-point reply to all the comments we received. The reviewers' comments are highlighted in red, while our reply in black.

Reply to Reviewer 1 - Anonymous

Comment 1:

This study first identified a dozen MTDs in the seismic record that shared one common sliding surface (H1). Then the authors attributed the climate characteristics such as greater swings of glacial-deglacial cycle, sea level etc. during the MPT as the cause of this common weak layer (H1). I have a major question on this interpretation: On the one hand the authors argued that because of the bigger swing of sea level during the MPT there were much more terrestrial sediment discharged into the study area and therefore produced the strata of coarser sediment and larger porosity, on the other hand the author also showed that the sedimentation rate during this same period was not higher! This seems a huge self-contradiction.

Reply 1:

Thank you for the comment. We have checked this issue on the sedimentation rate, and in the manuscript we stated that the sedimentation rate displays significant increase during glacial periods after the middle Pleistocene Transition. In the South China Sea, other studies have demonstrated the correlation between grain-size distributions, sediment supply, sea level variations and East Asian monsoon strength over the last 1.8 Ma using ODP wells 1144, 1145 and 1146 (Boulay et al., 2005, 2007). Our principal component analysis (PCA) also indicates the sedimentary factors (i.e. quartz, the total clay, smectite/(chlorite+illite), opal, coarse-sediment fraction, fine-sediment fraction, $\delta^{18}\text{O}$, sedimentation rate and porosity) are predominantly influenced by sea-level change and East Asian monsoon intensity. In general, three main phases of evolution have been recognized (see Boulay et al., 2007): 1.8 to 1.25 Ma, 1.25 to 0.9 Ma, and 0.9 Ma to present. In this latter period, Boulay et al. (2007) conclude that global sea-level changes control the variations in grain-size and the delivery of coarser sediments to the basin during sea-level lowstand intervals. As in this possible to see in the figure below (Figure R1), there is a clear overall increase in sediment supply after the MPT, and the first notable peak in sedimentation rate coincides with MIS16 (see the gray horizon bar in Figure R1), which is the first big drop of sea level after the MPT. There is also an increase in opal content (marked by the red star in the figure) and sand content, with distinctive peaks at each glacial intervals after MIS 16. Therefore, we observe that the overall increase in sediment supply and porosity (and of course opal and sand) starts after the MPT (so after MIS 16), and the absolute values are then modulated by glacial-interglacial cycles at 100 kyr time scale. We have completely rewritten this section of the manuscript to improve its clarity (see the details in Section *Lithological properties of ODP Site 1146*). Furthermore, we have also noticed that H1 (which corresponds to MIS 16) is not the only layer that can act as a sliding surface, but there are other four clearly visible horizons that we tied into the wells and coincide with glacial intervals. The new version of the paper includes this new result, and this is why we have also modified the title to underline this cyclicity in the formation of weak layers. Indeed, we are now able to prove the existence of multiple “weak layers” which are related to opal/sand rich intervals which are deposited because of climatic variations.

Figure R1 Correlation diagrams between d18O, opal, sedimentation rate and porosity derived from ODP 1146.

References:

- Boulay, S., Colin, C., Trentesaux, A., Clain, S., Liu, Z., Lauer-Leredde, C., 2007. Sedimentary responses to the Pleistocene climatic variations recorded in the South China Sea. *Quat. Res.* 68, 162–172. <https://doi.org/10/c3hpjp>
- Boulay, S., Colin, C., Trentesaux, A., Frank, N., Liu, Z., 2005. Sediment sources and East Asian monsoon intensity over the last 450 ky. *Mineralogical and geochemical investigations on South China Sea sediments. Palaeogeogr. Palaeoclimatol. Palaeoecol.* 228, 260–277. <https://doi.org/10.1016/j.palaeo.2005.06.005>

Comment 2:

The authors also said that the higher content of diatom and silicate could be used to explain the H1 layer's high porosity (and therefore weak), but did not elaborate. I recommend that authors dig a little deeper into why the break-up of diatom would increase the water content and porosity. It would be a much better contribution to our understanding the mechanisms of the common weaker layer, especially if it can be a bit quantified.

Reply 2:

Thank you for the comment, we agree that we did not elaborate enough our reasoning on the role of diatom/silica content in the formation of weak layers. In this new version of the manuscript, by using the data derived from ODP 1146, and in particular from the variation

of porosity along depth, we were able to preliminarily estimate the depth variation of undrained shear strength (S_u). (Figs.4F). The calculated S_u showed a distinctive drop at the interval of 96-100 mcd, which marks the first weak layer corresponding to the first “sudden” increase in opal content (occurring during MIS 16). Then, we conducted Ridge regression analysis to reveal the contribution weight of lithological features to the S_u . The results show that opal content and sedimentation rate are two dominated factors influencing the S_u (Table 1). See the detail in the sections of *Lithological properties of ODP site 1146*, *Geotechnical analysis results*, and *Ridge regression analysis for the dominant factors influencing the S_u* . Finally, we discussed how the opal content (or diatom content) controlled the weak layers based on the published literatures, which have investigated the influencing mechanisms through numerical simulation (Urlaub et al., 2015) and physical experiments (e.g. Shiwakoti et al., 2002), as well as the field geological investigation (e.g. Urlaub et al., 2018; Gales et al., 2023). See the details at the Line 300-312.

References:

- Gales, J.A., McKay, R.M., De Santis, L., Rebesco, M., Laberg, J.S., Shevenell, A.E., Harwood, D., Leckie, R.M., Kulhanek, D.K., King, M., Patterson, M., Lucchi, R.G., Kim, Sookwan, Kim, Sunghan, Dodd, J., Seidenstein, J., Prunella, C., Ferrante, G.M., 2023. Climate-controlled submarine landslides on the Antarctic continental margin. *Nat. Commun.* 14, 2714. <https://doi.org/10.1038/s41467-023-38240-y>
- Urlaub, M., Geersen, J., Krastel, S., Schwenk, T., 2018. Diatom ooze: Crucial for the generation of submarine mega-slides? *Geology* 46, 331–334. <https://doi.org/10.1130/g39892.1>
- Urlaub, M., Talling, P.J., Zervos, A., Masson, D., 2015. What causes large submarine landslides on low gradient (2°) continental slopes with slow (~ 0.15 m/kyr) sediment accumulation?: LARGE SUBMARINE LANDSLIDES ON LOW GRADIENTS. *J. Geophys. Res. Solid Earth* 120, 6722–6739. <https://doi.org/10.1002/2015jb012347>
- Shiwakoti, D.R., Tanaka, H., Tanaka, M., Locat, J., 2002. Influences of Diatom Microfossils on Engineering Properties of Soils. *Soils Found.* 42, 1–17. https://doi.org/10.3208/sandf.42.3_1

Reply to Reviewer 2 - Derek Sawyer

Comment 1:

This study aims to demonstrate that there is a widespread weak layer that facilitated several large submarine landslides to occur in the northern South China Sea and that this layer was deposited by changes resulting from the Mid-Pleistocene Transition. The concept/argument that a weak layer can be formed and would be related to climate-induced changes, is sound, and if definitively shown, would likely be significant and noteworthy.

Reply 1:

We thank Dr. Sawyer for highlighting the potential impact of our study.

Comment 2:

However, the primary areas of concern that I find after reviewing, are (1) there are not enough sufficient details in support of the claim regarding the weak layer. Abstract line 27-28 argue for a reduction in shear strength but it is not demonstrated to actually be weak with shear strength data, modeling, or other attempts at demonstrating this. There is circumstantial evidence regarding the potential role(s) of grain size, porosity, diatom presence, that the authors point to in support of defending a weak layer, but it is circumstantial evidence. There could be sufficient data for the authors to use in support of the claims, but in its present form, the paper needs additional details to support the claims about the layer being weak.

Reply 2:

Thank you for the comment. Due to the lack of a detailed mechanical characterization of the materials, we have preliminarily estimated the undrained shear strength (S_u) using a simplified approach which puts in relation void ratio (or equivalently porosity) and mineralogic content to the sediment shear strength. According to this and in agreement with critical state soil mechanics, for normally-consolidated materials, an increase in void ratio is associated with a decrease in the material shear strength. In other words, the less the microstructure is packed, the less is the material shear strength.

The variation of S_u along depth allowed us to quantitatively demonstrate a net reduction in shear strength for the sediments deposited after MIS16 (considered as the end of the MPT) and to prove the existence of “weak layers” formed during glacial intervals and associated with peaks in opal content.

We then used a statistical approach to prove that the opal content is the main parameter influencing the oscillations in void ratio (or equivalently porosity) of the sediments. This allows to state that variations in opal content are responsible of the variations of material strength: increasing opal content promotes the development of a weak layer.

Please see the details in the Line 215-230.

Comment 3:

The explanation of the role of H1 in the failure and the riming of failure, is very unclear.

Reply 3:

In the revised manuscript, we have added more investigation on the sedimentary changes across the H1. These can be summarized as it follows:

1. In section “*Seismic stratigraphy of the northern SCS and distribution of submarine landslides*” we highlighted the widespread and frequent occurrence of mass-transport

deposits above horizon H1, whilst mass-transport deposit is rarely observed in the strata below horizon H1.

2. In section “*Lithological properties of ODP site 1146*” we investigated the sedimentary variation across and above the horizon H1, such as coarsening of sediments, increased opal content, porosity and sedimentation rate, especially during the glacial periods.

3. In section “*Geotechnical analysis results*”, we calculated the variation along depth of undrained shear strength (S_u) based on the lithological parameters of Site 1146. The result shows that at horizon H1 the S_u rapidly decreased to a very low value, indicating the presence of “weak layer” around horizon H1. Meanwhile, the S_u above the H1 is much lower than that of the underlying strata, suggesting that the interval above H1 is weak and prone to have multiple weak layers.

4. In section “*Discussion*” we highlighted that H1 has an age of 0.63 ± 0.02 Ma (Fig. 5), corresponding to the end of MPT. This allows to put in evidence that the MPT drove the formation of weak layers at and above horizon H1. The results showed that the deep-sea deposition got into a stage that is prone to develop multiple weak layers with the accomplishment of the MPT, contributing to the widespread and recurrent mass-transport deposits above the H1 as observed on the seismic profiles.

Comment 4

The rationale that the layer must be weak because it underlies some slope failure deposits cannot be taken as fact and should be challenged for at least two reasons: (1) If the layer is actually a weak layer, how do the authors envision its role in the slope failures given that the horizon (H1) is apparently conformable and mappable across all the lines? The authors map it as a continuous surface from the unfailed area at ODP Site 1145 to beneath the MTD-1 landslide shown in Figure 2. Why would the weak layer be present beneath the landslides? The weak layer itself would be mobilized and eroded once the landslide occurs. I see in the figures and supplementary figures, that it does appear in some small areas that H1 is indeed missing and eroded, but it does not appear that this is common, but rather H1 is preserved.

Reply 4:

If we understand the comment correctly, the reviewer is suggesting that since horizon H1 is a conformable reflection from the un-failed area to beneath the landslide, then the reflection is probably not generated by the “weak layer”, as it should have been eroded/remobilized during the landslide motion. We agree that the sediment layer along which sediment motion occurs (so the weak layer) should have been partially, or entirely, eroded/remobilized during the downslope sediment motion, depending on the style of failure. However, the origin of this continuous reflection may, or may not, be related to the weak layer, as this will depend on its thickness compared to the vertical resolution (or frequency) of the seismic, as well as the contrast of impedance. Consequently, it is possible that the glide plane of a landslide is marked by a reflection that can be physically correlated with a reflection outside of the landslide, where the weak layer is still in place, that has the same signature.

A nice example is presented in the study published in *Geology* by Urlaub et al. (2018), from which the figure below is taken (Figure R2). The seismic line shows two slides (that the authors named *minor slide* and *major slide*) whose bases are marked by reflections that can be correlated to the un-failed area. We marked with a dashed violet line the seismic reflection corresponding to the glide plane of the *minor slide*, where the weak layer should be eroded, and with a continuous violet line the corresponding reflection in the un-failed section. For the *major slide* we used a dashed red line for the glide plane, and a continuous red line for the corresponding weak layer in the un-failed section. ODP 658 intersects the

glide plane of the *major slide*, and the weak layer of the *minor slide*, and quoting the authors: “The weak layer of the major slide has most likely vanished with the slide material. In contrast, the weak layers for the minor and buried slides are present in the ODP core because the site is located outside of their scar areas”. We provided two close-up views of these landslides (violet and red boxes), where we marked the glide planes by using arrows, and we also highlighted with an area of the same color the continuous reflection across the glide plane/weak layer. Now, it is possible to see that a glide plane (=eroded weak layer) is marked by a reflection that can be physically correlated to the un-failed area (= weak layer in place).

Figure R2 In this figure, modified from Urlaub et al. (2018), we show two examples of submarine landslides where the seismic reflection of their base has the same character below the landslide and in the un-failed area, even if below the landslide some material has been eroded, as testified by ODP 658.

In the new version of the paper, to improve the clarity of the text and to take into account the comment from the reviewer, we will call this horizon as glide plane when referring to the landslide, and as weak layer, when referring to the un-failed section in the well. We think that, at the ODP site 1146, the weak layer did not generate a visible reflection because either the contrast of impedance generated by the weak layer respect to the overlying and underlying sediment is too low to generate a reflection or because the resolution of the

seismic data at the depth of H1 is too large to resolve the weak layer. We think that in our study area a similar condition as in the paper from Wu et al. (2023), “Diagenetic priming of submarine landslides in ooze-rich Substrates”, published in *Geology*, may have occurred (Fig. R3).

Figure R3 The figure 3A of Wu et al. (2023), where horizon H2 is a seismically defined reflection that marks the base of seismic unit SU2, which is a landslide deposit.

In well data, H2 is an ~13-m-thick zone interpreted as an overcompacted and low permeability unit, which covers a layer with low-density, low-velocity, and high-porosity responses, and these sharp changes in petrophysical properties resulted in H2 being expressed by a high-amplitude, positive-polarity seismic reflection. In addition, the well data show another petrophysically distinct interval and the authors named Hs, which is ~5 m thick lie immediately above H2. Hs is revealed only in the well data, and does not have a corresponding seismic reflection. Hs is characterized by relatively high water content and void ratio (and hence low shear strength and high compressibility) and a low acoustic velocity, and authors interpret it as a weak layer (Fig. R4).

So in essence we think that horizon H1 in our study also includes the weak layer outside of the landslide deposit as the weak layer does not have its own seismic signature.

Figure R4 The figure 4E of Wu et al. (2023), displaying the variation of Vph, density, water content, and void ratio with depth at ODP site 762

References:

Urlaub, M., Geersen, J., Krastel, S., Schwenk, T., 2018. Diatom ooze: Crucial for the generation of submarine mega-slides? *Geology* 46, 331–334. <https://doi.org/10.1130/g39892.1>
 Wu, N., Steventon, M.J., Zhong, G., 2023. Diagenetic priming of submarine landslides in ooze-rich substrates. *Geology* 51.

Comment 5

When a submarine landslides occurs, the mass moves out of the source area and eventually the mass is deposited (MTD) on the seafloor some distance from the source area, unless they are localized slumps that do not move very far from the source area. It seems the data support that former, that these are slides that mobilized away from where they originated. If true, the weak layer that the mass failed along is not the one the deposit is resting on, but rather the scarp area where the landslide originated. Do you have the data to show this, that the detachment in a scar area is H1?

Reply 5

In general, MTDs overlying the paleo-seafloor would display gradual thinning thickness and pinch out towards their margins due to the un-confinement. Meanwhile, the subsequent sediment would display onlapping upon the top surfaces of MTDs.

However, the MTDs in the study area display very different features, which are characterized by sharp interface with the adjacent undeformed strata, without gradually thinning thickness and onlapping upon the tops (Fig. 3; Figs. S2-S7). They display similar features to frontally confined MTDs, which generally move along a buried surface, but rather than the seafloor (Frey-Martínez et al., 2006).

The figure below, which is taken from the pre-print of Sager et al. “Assessment of Submarine Landslide Volume (DOI: <https://doi.org/10.21203/rs.3.rs-3205387/v1>), helps explaining how we interpreted horizon H1 (and associated weak layer) relative to the landslide deposit above it.

The figure shows a seismic profile across the Ana slide. Contorted to chaotic/transparent seismic reflections which are diagnostic of landslide deposits are visible either close to the escarpment (named V_{e_r} , see the green box) or in a more distant location (named $V_{a_{bulk}}$, see the black box). In both cases, the deposits occur above the same horizon (marked in yellow proximally and in green distally). V_{e_r} , marked in brown colour and defined as the *remaining volume*, is defined by the authors as the volume of undifferentiated landslide material that remained inside and was mobilized, whereas $V_{a_{bulk}}$, marked by a grid, is defined as the *bulk accumulated volume* and is characterised by chaotic, transparent, and disrupted seismic facies representing mobilized and affected landslide material and slope sediment.

In our study, the landslide deposits above H1 can be related either to V_{e_r} or to $V_{a_{bulk}}$, as we cannot always correlate the landslide with the respective escarpment, and horizon H1 is the equivalent of the basal shear surface as in the figure above.

In the revised manuscript, we have described these features in the *Results (Line 148-156)*, supporting the idea that the slide surface beneath the present MTDs was a weak layer before failure.

References:

Frey-Martínez, J., Cartwright, J., James, D., 2006. *Frontally confined versus frontally emergent submarine landslides: A 3D seismic characterisation. Mar. Pet. Geol. 23, 585–604.*
<https://doi.org/10.1016/j.marpetgeo.2006.04.002>

Comment 6

If that's the case, then it begs another questions...because the seismic data show that the moving mass came to rest on H1, it is puzzling to understand what it suggests about the timing of the failure. If H1 was the seafloor at the time of failure, what is the mass itself composed of? I believe the authors envision a scenario in which H1 was deposited during the MPT, and then some time later slope failures occurred and detached along H1, which would have been now at some depth below the seafloor. It's not clear to me how this is explained, but it really needs to be examined and presented for a more convincing analysis and understanding of these landslides and the role of H1.

Reply 6

H1 was not at the seafloor at the time of failure. Our model is that H1 was deposited at around MIS16, then other sediments accumulated above H1, until the landslide occurred and detached along H1 moving all sediments above it. It is worth nothing that we do not have constraints on the timing of the landslide.

As we mentioned in the Reply 5, we highlight that the MTDs in the study area can be defined as the frontally confined MTDs introduced by Frey-Martínez et al. (2006) and thus the submarine landslides moved along the weak layers, and not along the seafloor, which does not make sense (see the model/interpretation of the figure above in Reply 5, which clearly explains our interpretation). In addition, H1 is visible at the base of the landslide deposit and laterally extends into the undeformed strata, indicating that H1 was formed before the occurrence of the landslide. We have added more work/analysis (e.g., undrained shear strength; Lines 190-215) to explain this and strengthen our argument that H1 serves as a weak layer. Please also see our Reply 2 and Reply 3.

References:

Frey-Martínez, J., Cartwright, J., James, D., 2006. *Frontally confined versus frontally emergent submarine landslides: A 3D seismic characterisation. Mar. Pet. Geol. 23, 585–604.*
<https://doi.org/10.1016/j.marpetgeo.2006.04.002>

Comment 7

Another possibility that is not addressed is perhaps the layer itself is not weak, but somehow causes a weakening of the strength characteristics of the layers immediately above it? These specifics of what the data show and what is being interpreted really need to be addressed more robustly. There is no description of any shear strength data at Site 1145 or for Leg 184.

Reply 7

The variation of porosity along depth, observed at site 1146, puts in evidence a clear reduction of porosity in correspondence of H1. As is also discussed in Reply 2 and 3, sediment shear strength is related with its porosity: a reduction in porosity implies a reduction in shear strength.

However, this does not allow to exclude a (transient) weakening of the strength of the layer above it. In fact, during the failure of the weak layer, excess pore water pressure may have developed and induced a consequent transient upward seepage in the layer directly above the weak layer. This is expected to cause a temporary reduction in effective stresses and a temporary increase in void ratio. This implies a temporary reduction in shear strength. This effect however is expected to end after the end of the transient seepage process, after which overlying layers are expected to regain their strength.

Comment 8

Based on my review of the ODP database, there unfortunately appears to be no shear strength measurements acquired. This is unfortunate as shear strength is usually a standard measurement collected. The authors need to present this information and try to work around this particular challenge, if you want to stick with the claim that it is in fact a weak layer. Ideas that come to mind would be to request core samples across the H1 surface. The cores are presumably too old to offer much hope for direct shear strength measurements, but you would do other laboratory studies to address the shear strength characteristics of that horizon.

Reply 8

That is correct. No shear strength measurements have been acquired during the drilling, and we agree that the cores are probably too old to perform any reliable mechanical characterization. For this reason we did not ask samples from the ODP office.

However, in the current version of the paper, starting from the porosity and mineralogic content, we included an estimation of the undrained shear strength at ODP site 1146. The calculated results show that H1 corresponds to a local minimum of undrained shear strength, suggesting that it is a weak plane. In addition to that, above H1 shear strength is generally low and characterized by multiple local minimums, suggesting the presence of multiple weak planes. "Please also see Reply 2 and Reply 3.

Comment 9

If you want to argue that diatoms are breaking and causing pore water pressures to spike (and therefore weakening the layer), you could examine the diatoms under microscope to address that possibility. There could also be a modeling effort to assess slope stability of these materials and these slopes.

Reply 9

We believe that the marked lithological contrast between the opal-rich layer (low density high porosity/high water content) and the overlying clay-rich layer (high density and low porosity/low water content) is enough to generate a weak layer, as demonstrated by the S_u values. However, it is also possible that some overpressure may exist due to the water released from the diatoms. We did not include this mechanism as in the literature, see for example Volpi et al., (2003), this mechanism is generally considered when the opal content is >40%, and since we have much lower values we do not really know if this can be applied.

References:

Volpi, V., Camerlenghi, A., Hillenbrand, C.-D., Rebesco, M., Ivaldi, R., 2003. Effects of biogenic silica on sediment compaction and slope stability on the Pacific margin of the Antarctic Peninsula: Effects of biogenic silica. *Basin Res.* 15, 339–363. <https://doi.org/10.1046/j.1365-2117.2003.00210.x>

Comment 10

There is also insufficient evidence about the line-to-line correlations that the H1 horizon is indeed the same across all the lines. How were these done?

Reply 10

In the revised manuscript, we have expanded our study area that is covered by 24 two-dimensional (2D) multi-channel seismic lines, with a total length of 2236 km, with a grid of 8×16 km. The horizon picking is performed making loops across each seismic profile, so that the final interpretation is consistent across the entire dataset. Furthermore, the intersected seismic profiles and the associated interpretation clearly show that H1 is a regionally continuous horizon that could trace across all the seismic lines (Fig. R5). Meanwhile, based on the seismic-well tie and depth-age models at Site 1145 and 1146, we could obtain the ages of the critical seismic horizons (Fig. 2 and Fig. S2). The results at these two different sites are consistent, further supporting the validation of our interpretation.

Figure R5 The intersected seismic profiles in the study area.

Comment 11:

Where are the sonic and bulk density logs coming from? Why do you need to model the P-wave velocity and elastic parameters when you have the sonic and bulk density?

Reply 11:

The bulk density data is the laboratory measured data from ODP Site 1145 report. However, there is no sonic log for the Site 1145, and the valid laboratory-measured sonic data is only for the upper 58 meters composite depth (mcd). Therefore, we need to model the P-wave velocity for making the synthetic seismogram of ODP Site 1145 (*See the detail in the Supplementary*).

Site 1146, located ~20 km to west of the study area (Fig. 1B), has completed log data and lithological data. Therefore, the revised version mainly used data from this site to define the depth and ages of the critical seismic horizons in the landslide-dominated area. Furthermore, the 1146-based results are consistent with the 1145-based results, which further support the credibility of our P-wave velocity inversion for ODP site 1145, seismic-well tie, and seismic interpretation.

Comment 12:

What wave are you convolving with the reflection coefficient series?

Reply 12:

We used 77 Hz, zero-phase Ricker wavelet to convolve with the reflection coefficient series. The dominated frequency of 77 Hz is selected based on the spectrum analysis of the seismic profiles for the shallow interval.

Comment 13:

It is hard to understand the first paragraph in the Results Section. Particularly in Lines 86-89, it seems incomplete: “For the other five MTDs that did not shared the basal surface of H1, four of them (MTD8, 10, 11 and 12) superimposed on the older MTDs (MTD1, MTD4 and MTD5) that had basal surface of H1 (Figs. 1C, S7 and S8).” Please explain what you mean by “superimposed on” and what is the significance/relevance of this?

Reply 13:

We agree that this section is confused, and we have deleted it. Horizon H1 is not the sole “weak layer”, and there are multiple horizons that can be tied with opal-rich layers formed during glacial intervals. We have completely revised the manuscript to account for this new fundamental aspect in this revised version.

Comment 14:

If the H1 surface can be shown to be a weak layer, what then was the triggering mechanism(s) and when did the failure(s) occur?

Reply 14:

This study is focused on the formation of weak layers due to climate transition. Though the triggers of slope failures are beyond the aim of this study, we provided some possible triggers in Lines 316-318. We think that earthquakes may be the main triggers for the occurrences of multiple slope failures.

Comment 15:

It appears to me that in the seismic data of Figure 2 that H1 at Site 1145 has a lower reflection coefficient and therefore amplitude than it does as mapped beneath MTD1. It would be good to explain or present reasons for what is controlling this amplitude response.

Reply 15:

This may be caused by the lateral variation caused by the impedance difference across the weak layer and the glide plane beneath the MTDs. When compared to the hemipelagites/ fine-grained turbidites/sheeted contourites, MTDs are generally dominated by lower porosity, higher density, and higher acoustic velocity Dugan (2012). As a result, the impedance difference between the MTD and the underlying strata would be greater, causing the seismic amplitude beneath the MTD is higher than the corresponding surface shown at unfailed areas.

References:

Dugan, B., 2012. *Petrophysical and consolidation behavior of mass transport deposits from the northern Gulf of Mexico, IODP Expedition 308. Mar. Geol.* 315–318, 98–107.
<https://doi.org/10.1016/j.margeo.2012.05.001>

Reply to Reviewer 3 - Davide Gamboa

Comment 1

The manuscript by Wang and co-authors presents further results on a compelling aspect on precursors on submarine mass-wasting and their relationship to climate. This is a compelling but challenging topic to investigate, although recent works have shed further insights on it (e.g. Murlaub et al, 2018). The intent is to correlate changes in Earth's climate with sediment deposition patterns and characteristics, with a focus on bioproductivity and consequent bioclastic input to submarine basins. This is a further contribution on an important open scientific question, particularly at the current research scenario where efforts are being made to understand the influence of climate change on different natural geohazards.

The main premise of the work is to provide further proof on the effect of bioclastic elements on the location of weak layers and intervals, namely after the Middle Pleistocene Transition (MPT). The results are plausible and presented through the integration of multi-scale observations from seismic and IODP data to support the premise.

Reply 1:

We are grateful to the reviewer for the positive comment and for considering our study of interest to the broad community of earth scientists studying the effect of climate change on natural hazards.

Comment 2:

Although the main idea is broadly explained in the manuscripts current form, it feels it may be further extended to provide a more valuable and impactful work. There is quite a bit of room to expand both text and figures within the journal format, and that should be explored. For example, some of the figures currently in the supplementary material could be part of the main manuscript. A relative sea-level curve, often included on such studies, could be presented as well for a better correlation with the climatic shifts the authors focus on, and would complement the descriptions on the text.

Reply 2:

Following your suggestion, we have expanded the text and figures for supporting our idea. In the revised version, we have added the sections on geotechnical analysis, ridge regression analysis, and principal component analysis (PCA). We have also modified existing figures and added new ones showing sedimentary parameters variation with depth (Fig. 4) and time (Fig. 5), effective stress-void ratio plot (Fig. 6), loading plot of principal components (Fig. 7), principal components (PC1 and PC2) variation with time (Fig. 8), as well as the conceptual model in Fig. 9. The relative sea-level curve has been added in Fig. 5. Moreover, more seismic profiles showing the widespread post-MPT submarine landslides are displayed in the Supplementary (Figs. S2-S7).

Comment 3:

The authors could also test if the premise of the climate-derived failure precursor represented by H1 is also valid for other common detachment levels of other MTDs, as for example H3. Although I understand that the main premise is the shift in conditions observed in the vicinity of H1, it would still be of interest to assess if posterior high productivity episodes occurred at earlier times and if they had the same influence.

Reply 3:

We thank the reviewer for this excellent comment. In the first version of the manuscript, we primarily focused on the role of horizon H1 as a weak layer and how its formation can be linked to the MPT. Following your comment, we have expanded our study (this is why the revision took one year) and we discovered that the existences of multiple horizons over which landslide deposits accumulate. We tied these horizons in the ODP Site 1146, and we found that they correspond to intervals rich in opal that formed during lowstands of sea level. The first horizon H1 corresponds to MIS 16, which is the first great drop in sea level after the MPT. Following this discovery, we have completely revised the manuscript to present this cyclicity in the formation of weak layers, and consequently suggested a new title for the manuscript.

Comment 4:

I take the chance to raise a point, which may or may not be followed, regarding the detachments. The widespread use is to refer to a weak layer, but questions had been raised if using a detachment/basal interval would be more appropriate, especially since the vertical heterogeneity between alterations of fine, coarse and bioclastic material influence the failure. On the seismic resolution here presented the detachment is generally limited to a single reflection/surface, but since there is higher resolution data available from the borehole is it possible to have further insights on this? An aspect that can be further expanded is how these changes effectively lead to higher proneness of failure.

Reply 4:

Weak layer is generally regarded as a preferential failure plane of submarine landslides, which could exist at the unfailed areas and imply the risks for future slope failure (Gatter et al., 2021). In contrast, the detachment/basal intervals are used to indicate to the bottom of mass transport deposits, which are the results of submarine landslide. Therefore, detachment/basal interval would not be present at unfailed areas, and could not be used to estimate the risk potential of future landslides. In order to well assess the risk potential through estimating the location, distribution and volume of future slope failures, we prefer to use “weak layer” at the areas outside landslides, and call the corresponding layer as glide plane at landslide areas.

In the revised manuscript, we firstly identify the horizons H0-H5 on seismic profiles (Line 108-124), and then investigate the lithological variation across the horizons at site 1146 (Line 158-188). Based on the seismic-reflector and lithological features, we discussed the relationship between lithological changes driven by climate transition with the distribution and recurrence of slope failures (Section *Discussion*).

References:

Gatter, R., Clare, M.A., Kuhlmann, J., Huhn, K., 2021. Characterisation of weak layers, physical controls on their global distribution and their role in submarine landslide formation. *Earth-Sci. Rev.* 223, 103845. <https://doi.org/10/gm9857>

Comment 5:

There are mentions to previous works on the aspect of shell breakdown and dewatering but seems to be limited reference to what is available for this study that could attest for it.

Reply 5:

Thanks to the additional data we collected while doing the revision, we can provide a better constrained interpretation for the formation of weak layers. In the revised manuscript, we discuss the role that the marked lithological contrast between opal-rich (low density high porosity/high water content) and the overlying clay-rich layers (high density and low

porosity) plays for generating a “weak layer”, and we introduced the undrained shear strength to show it. Although it is possible that some overpressure may still form due to the water realised from the breakdown of the diatoms, we do not think this process is relevant due to the overall low percentage of opal. Indeed, Volpi et al. (2003) had shown that water release from diatoms breakdown generates overpressure when the opal content is >40%, and this value is much higher than the opal content in ODP Site 1146.

References:

Volpi, V., Camerlenghi, A., Hillenbrand, C.-D., Rebesco, M., Ivaldi, R., 2003. *Effects of biogenic silica on sediment compaction and slope stability on the Pacific margin of the Antarctic Peninsula: Effects of biogenic silica. Basin Res. 15, 339–363. <https://doi.org/10.1046/j.1365-2117.2003.00210.x>*

Comment 6:

Another discussion point that could be pursued is how the high opal levels see to be offset from H1. Are there any implications from this in terms of failure inception. i.e., is the weak “layer” at the high diatom level or below? As there is well data available, perhaps some aspects on geotechnical properties at these specific weak layers/intervals could be expanded, if published or available?

Reply 6:

The seismic-well tie result shows that at Site 1146 the seismic horizon H1 has a buried depth of 96-100 mcd (Fig. 4). At such depth, the undrained shear strength presents a local minimum (Fig. 4), implying that the horizon H1 could be regarded as a weak layer. Based on the previous studies on the influence of diatom ooze on the sediment geotechnical features, in *Discussion* section we explained how the weak layer was formed. Please see also the replies to Reviewer 2.

Comment 7:

As a final consideration, aspects of the discussion could focus on more margins affected by mass-wasting contemporaneous with the MPT. This is, after all, the premise that the reader gets from the title of the submission. Even if other examples do not attest for identical preconditioning conditions, it will be worth summarizing which other climate-driven factors can influence widespread landslides.

Reply 7:

Following your suggestion, we have added more examples on the weak layer formation related to MPT in the revised manuscript (Line 319-329). Besides, we have also added an example on the relationship between landslides and other climate changes and the associated sea-level fluctuations on the Antarctic continental margin (Line 329-332).

These examples together implied that climate change plays a crucial role in the weak-layer formation, mainly through influencing sediment types and sedimentation rates, which would control the formation and frequency of weak layers.

Dear Editor,

Please find below a point-by-point reply to the comments from the three reviewers.
For clarity, the comments are in red while our replies in black.

**Reply to comments of Reviewer 1**

1) The authors made substantial changes in the revised ms, one of which is to
recognize H1 to H5 as weak layers, in contrasting to the claim of H1 as the only and
common weak layer in the first submission. Additional data were also added and more
analyzing work were provided. Those additional work sufficiently answered my two
questions: 1) it clarified the original concern of "large sediment supply but non-higher
sedimentation rate" contradiction; and 2) the authors decided not to further elaborate
the diatom break-up issue other than citing published papers, therefore it becomes a
moot point.

1) We are grateful to the reviewer for positively evaluating the revised manuscript and
the additional work we introduced to provide a reply to the two main questions.
Regarding point 2 of the comment, since this revision, we have further expanded the
section about how diatoms may influence shear strength also providing a short review
of the current lines of thought. Additionally, we better elaborated the discussion about
how low amounts of diatoms (<10%) may reduce shear strength by controlling
porosity in order to clarify our numerical approach.

2) It's worth pointing out that the issue of "whether the weak layer can be preserved in
the seismic record after a landslide gliding over it" should be fully explained by the
authors. That's beyond my knowledge base so I will have the other reviewers to judge
whether they are satisfied with the authors' arguments in the revised ms.

2) In the new version, we have considered this comment and provided a better
explanation to what happens to a weak layer after the emplacement of a submarine
landslide. As we discussed in the previous reply, we agree that the sediment layer
along which sediment motion occurs (so the weak layer) should have been partially,
or entirely, eroded/remobilized during the downslope sediment motion, depending on
the style of failure. However, at seismic scale, the base of a submarine landslide
deposit is often marked by a continuous seismic reflection that can be physically
correlated with a seismic reflection of equal characteristic outside of the landslide in
the un-failed area, where the weak layer is still in place. This means that the
continuous reflection at the base of a landslide is not solely generated by the weak
layer, as this will depend on its thickness compared to the vertical resolution (or
frequency) of the seismic, as well as the contrast of impedance.

Reply to comments of Reviewer 2

1) As was noted in the previous review and acknowledged by the authors, the actual strength of these horizons was not measured. The authors have now applied a numerical model to simulate the undrained shear strength. However, there are two issues with this technique and the conclusions drawn from it. The higher porosity zone is equated to a lower undrained shear strength. This is not necessarily true for opal/diatom bearing horizons.

1) Given the low contribution of opal to the sediments (opal content is between 1 and 7% in mass and the similar density to the clastic sediments, the standard approaches used in soil mechanics could be used to estimate the undrained shear strength.

In order to obtain the along-depth variation of the sediment shear strength, we have analyzed the shear strength for 59 samples of ODP Site 1146 requested from the Kochi Core Center in Japan in 2024, and the depths of these samples range from 0.4 to 137 mcd. Given these samples are old and disturbed, we cannot directly measure the shear strength. However, we could adopt the well-known approach introduced by Skempton (1957), which suggested that the undrained shear strength (S_u) could be preliminarily estimated from the plasticity index (PI , defined as the difference of limit and plastic limit) and the in situ vertical effective stress (σ'_v) for normally consolidated clays. According to this approach:

$$\frac{S_u}{\sigma'_v} = 0.11 + 0.037PI \quad (1)$$

This ratio is particularly useful for identifying variations in soil strength attributable to changes in soil composition.

The PI could be obtained through measuring the plastic and liquid limits for the 59 samples of ODP Site 1146 through cone penetration method, following the standard for geotechnical testing method (GB/T 50123-2019).

The experimental data clearly show local minimum of S_u/σ'_v at the depths of SH1-SH5 (horizontal lines), suggesting the presence of weak layers at those depths (Fig. 4C). These layers also display higher porosity values.

We have added the more detailed explanation in the revised version.

2) If opal is coming from diatoms, diatoms are known to preserve high porosity while at the same time having higher shear strengths, not lower. This has been detailed by Wiemer and Kopf, 2017 and Shiwakoti et al., 2002. This Wiemer and Kopf paper is not referenced in this manuscript and the Shiwakoti reference is listed but only vaguely saying that opal may influence geotechnical properties (Lines 300-302). Right now, the paper is trying to show weak layers by connecting it to opal, but the literature shows, it's not that simple, and actually suggest it enhances the strength. The rough surface area of the diatom shell (frustule) contributes to high shear strength. At the same time though, the frustule has high intraparticle porosity.

2) Many authors observed that the shear strength of soil containing diatom increases
with the diatom content due to the interlocking of the diatom particles; see the studies
from: Shiwakoti et al, 2002; Díaz-Rodríguez, 2011; Tanaka et al., 2012;
Díaz-Rodríguez & Moreno-Arriaga, 2017; Wiemer & Kopf, 2017; Zuluaga et al.,
2021; Zuluaga et al., 2022.

However, **for low content of diatoms (<25%)**, the mechanical behavior of
soil-diatom mixtures in terms of shear strength, compressibility, and index properties
is practically **unaffected** by the presence of diatoms (Díaz-Rodríguez, 2011; Tanaka
et al, 2012; Zuluaga et al.; 2022; Zhang et al, 2023). According to Tanaka et al. (2003),
when a small quantity of diatoms is mixed in the soil matrix, neighboring diatom
particles do not interact with each other and the previously cited interlocking cannot
take place.

In the case presented in our paper, the maximum observed diatom content is
approximately 7% (see Fig. 4B). This suggests, according to the literature cited above,
that diatoms **do not interact** among each other and thus **do not increase** shear
strength. Consequently, the low diatom content allows us to estimate the mechanical
strength of soil-diatom mixtures by following the standard theories of soil mechanics,
such as the one based on the critical state soil mechanics (Schofield & Wroth 1968),
according to which increases in porosities (or, equivalently, of void ratios) are
associated with a reduction in shear strength.

Diatom particles are characterized by a porous microstructure. Therefore, it is
necessary to distinguish between the voids within the diatom particles
(“microporosity”) and the voids between the particles (what is usually described as
porosity). The particle density (not including by definition the interparticle voids) can
be used to estimate the importance of microporosity on the properties of the
soil-diatom mixture. According to the data available in the literature, diatom particle
density ranges between 2.2 and 2.4 t/m³ (Hamilton, 1976; Shiwakoti et al., 2002;
Tanaka et al., 2012; Wiemer & Kopf, 2017; Zuluaga et al., 2021; Zhang et al., 2023).
In the case presented in our paper, the average particle density (including both diatom
fossils and soil particles) ranges between 2.6 and 2.75 t/m³ (see Fig. 4B). These values
are very similar to those expected for a soil without diatoms (Hamilton, 1976),
suggesting a limited influence of micropores on the porosity.

As discussed in Díaz-Rodríguez & Moreno-Arriaga, 2017; Wiemer and Kopf, 2017;
Zuluaga et al., 2021; Zhang et al., 2023, diatoms also induce an increase in the size of
the intergranular voids. This increase in the (macro) porosity is related to a reduction
in undrained shear strength of the soil-diatom mixture, as discussed in our paper and
further detailed below. This conclusion, however, holds only when the diatom content
is sufficiently low so that diatom particles cannot interlock between each other, which
is exactly our case. Consequently, we have a mechanistic explanation of why the
accumulation of diatoms during glacial intervals promoted the formation of weak
layers, and so the main issue highlighted by the reviewer is addressed.

We have included the reasoning of the opal-bearing weak layers in the revised

manuscript.

3) It's not explained where the opal is coming from at Site 1146. Is this due primarily
diatoms, primarily not diatoms, or some mixture of other biogenic silica (opal)
sources?

3) Wang et al. (2003) presented a detailed investigation of the opal accumulating on
the northern South China Sea margin, based on data from ODP Site 1146. Their
results show that the opal is mainly composed of radiolarians and diatoms, and much
lower amounts of sponges (see the following Figure R1 from Wang et al., 2003). We
will include this information in the revised manuscript.

Figure R1: Correlation of the oxygen isotope record with sea-level fluctuations and
accumulation rates of radiolarian, diatom and sponge at ODP Site 1146 (Modified
from Wang et al., 2003).

4) This is also not well-defended to define “coarse” fraction as 19 microns. Typically,
the coarse fraction would be defined as 63 microns and above (i.e. sand-sized and
coarser). Therefore, it is possible to have high-porosity and high shear strength in
diatom-bearing horizons and at the same time being in the “coarse” fraction if one
considers 19 microns as coarse.

4) We agree with the reviewer that the “coarse” fraction was poorly defined in the
paper. The definition of the ‘coarse’ fraction as 19 μm is from Boulay et al. (2007)
who conducted end-member modeling and divided the terrigenous sediments at site

ODP 1146 into three end members at 4 μm , 9 μm , and 19 μm sediment fractions.
Accordingly, they defined the 19 μm and 4 μm end-member as a coarse and fine
fraction, respectively. Our previous version did not introduce the end members of the
sediments, which probably confused the reviewer. We will use the ‘coarse
end-member’ and ‘fine end-member’ to replace the previous ‘coarse fraction’ and
‘fine fraction’, respectively, in the new revised version. The increase of ‘coarse
end-member’ is mainly used to support the glacial sediment supply increase after
MPT. We have included this information in the revised version.

Reply to comments of Reviewer 3

1) The revised version of the manuscript by Wand and co-authors addressed the
previous comments by reviewers, and added new data and results that, in my opinion,
greatly improved the scope of the link between climate-influenced sedimentation, its
role in weak layer development and geohazards. I have a few additional comments on
this version which mostly consist on some clarifications.

1) We are grateful to the reviewer for positively evaluating our revised manuscript.

2) Title: In my opinion, the title lacks an element to identify the focus on marine
environments.

2) We agree, and we have added the word “submarine” to clarify the focus on the
marine environment.

3) There are some typos or words missing here and there. This can be fixed with a
thorough proof read.

3) Thank you for checking. We have corrected the typos in this new version.

4) Line 76: maybe I’m being picky, but can the statement that “show how the
occurrence of submarine landslides in the Quaternary northern SCS can be linked to
the climatic variations” be miss-interpreted as the climate variation being the
cause/trigger for the landslide? The sentence is valid, but can it be stated in a less
ambiguous way?

4) We have modified the sentence to avoid any ambiguity between triggers and
preconditioning factors.

5) Line 100: As sediment input and type is a major, or the major element of this paper,
this section needs to expand on the other sediment types relevant to the work, namely

the siliceous ones that tend to be biogenic. There is also a lack of information of the
source of opal, as it may come from different organisms that develop in different
climatic settings – this may link to the aspects that authors further highlight regarding
glaciations and monsoons.

5) In the revised manuscript, we have included detailed information on the origin of
the opal referring to the work by Wang et al. (2003) who analyzed all siliceous species
in the ODP site 1146. We have used these new data to further elaborate on the role of
climate and sea level change in controlling sediment supply and opal content.

6) Line 148: The title and introduction point to the submarine mass-wasting as a
major element of the paper, but mention to their overall characteristics ends up being
limited. I am not saying that an exhaustive and boring description of them should be
made, but since they are the major feature and occur due to the weak layers analyzed
there should be more on the mtd relevance to show to the reader. For example, in the
body of the text the link of the MTDs with the horizons is limited to “Horizons H1 to
H5 represent the basal glide planes of multiple MTDs”, with then the specific number
just for H1. If the cyclicity is important, its link to the several the opal-rich horizons
deserves more focus. For example, the largest MTD by far is linked to H2 and H3.
That is relevant, but the text only highlights H1 as significant. This type of really
relevant observations could be in this point/paragraph. I think the description of the
seismic facies is secondary for this level of journal and only take up space where the
points above can be developed. The facies can be described in the supplementary
material.

6) Agreed. In the revised version, we have provided more details on the MTD
characteristics and related horizons, and move specific seismic facies description in
the supplementary.

7) Line 190 to discussion: The specific section could start with a simple sentence with
the purpose of the analysis. For example, why make the Geotechnical analysis?
Picking up on your reply to Rev 2, it was a way to confirm/further support the
presence of the weak layers. We only get that link at the end with the mention to H1,
H2 an H5. Can it mean H3 and H4 risk being “not weak”? Does it have implications
of the paper scope? H3 also seems to be at a (local?) minimum, so why not mention
it?

7) In the manuscript we wrote that “*other intervals with minimum in S_u are observed*
*at the depths of horizons H2 to H5*”, and with this sentence we wanted to highlight
low S_u values are also present at all the horizons H2, H3, H4 and H5, not only H1.
Since we understand this sentence might be misunderstood, we have modified it in the
revised manuscript. In addition, in this new version we have clarified why the
geotechnical analysis is needed to support the presence of weak layers at the
beginning of the section *Geotechnical analysis results*.

8) It is not too clear what the principal components want to represent. The reader can
understand the results, but clarify the main purpose to do it. In line 240, does it make
more sense to mention the fine sediment, which has the lowest of all values, or the
clay component?

The interpretation of the PCA (or at least part of it) and relationship with the sea level
changes is repeated in the discussion, so the results section can limit itself to show and
explain the data. Maybe some of the expanded interpreted PCA made in the results
can be moved to the discussion, if relevant.

8) Agreed. We have added one sentence to show our aims of conducting PCA. See
Line 252-253. Also, we have shortened the section of principal component analysis
and interpretation and moved part of previous explanation to the current Discussion
section.

9) Line 314: “promoted the formation of multiple weak layers (i.e. H1-H5), which can
explain why MTDs are primarily, if not uniquely, observed in late Quaternary
sediments”. I don’t disagree with this, but maybe this is an additional factor and not
the main one. It would be worth referring to the higher rates of landslide occurrence
during sea-level regressions, if amplitude of sl changes influence this as there were
changes before the MPT but apparently less significant(?) and possibly with other
margin sedimentary regimes? The sed supply rate would be worth mentioning pre and
post-MPT could be worth mentioning.

9) Higher amplitude sea level changes can act as a trigger mechanism and definitively
be a reason for the higher number of landslides in post MPT sediments. Although we
can only speculate on the role of sea level, we followed the reviewer’s comments,
expanding the discussion to account for the role of higher amplitude sea level changes
and associated increase in sediment supply in the variation of landslide occurrence
across the MPT.

10) Line 325 to 329: at first, I wondered why suddenly carbonates and volcanoclastic
material are relevant for this, but they do link to the cyclicity argument. My
suggestion is to 1) keep it, adding the argument of relevance of this cyclicity for
failure preconditioning in other settings, and 2) move it to after the Antarctic case.
The question is, is mentioning the carbonate/volcanic examples, would it be worth
also looking for other examples where cyclicity and slides have been observed, to
provide a higher global impact of the results? Or keep the focus on the silica/opal
examples?

10) Thanks for the comment. We have revised this section to better highlight that
climate could control the cyclic formation of weak layers not only in clastic settings
but also in carbonate margins. We discuss two examples describing post-MPT
sediments and one that spans also to the Miocene. At the best of our knowlodge, we did

not find other examples where the lithology and chronology of weak layers have been
quantified (see also Table 1 in Gatter et al. 2021) to further expand the discussion
about the cyclicity of weak layer formation. This indeed supports the novelty of our
findings that may foster future research towards this direction.

**References cited**

Boulay, S., Colin, C., Trentesaux, A., Clain, S., Liu, Z., Lauer-Leredde, C., 2007.
Sedimentary responses to the Pleistocene climatic variations recorded in the
South China Sea. *Quat. res.* 68, 162–172.

Díaz-Rodríguez, J.A., 2011. Comportamiento monotónico de suelos diatomáceos. In
*Revista Investigación de Desastres Naturales, Accidentes e Infraestructura Civil*;
Universidad de Puerto Rico, Mayagüez, Puerto Rico., Volme 12, pp. 27–34.

Díaz-Rodríguez, J. A., Moreno-Arriaga, A., 2017. Contributions of diatom
microfossils to soil compressibility. In *Proceedings of the 19th international*
*conference on soil mechanics and geotechnical engineering* (pp. 349-352). Seoul,
Korea: Korean Geotechnical Society.

Hamilton, E. L., 1976. Variations of Density and Porosity with Depth in Deep-sea
Sediments. *Journal of Sedimentary Petrology* 46 (2), 280-300.

Koumoto, T., Houlby, G.T., 2001. Theory and practice of the fall cone test.
*Géotechnique* 51, 701–712.

Nova, R., 2013. Soil mechanics. The McGraw-Hill Compagines, S. r. l. Milano.

Schofield, A., Wroth, C., 1968. *Critical State Soil Mechanics*. The McGraw-Hill,
London.

Shiwakoti, D.R., Tanaka, H., Tanaka, M., Locat, J., 2002. Influences of Diatom
Microfossils on Engineering Properties of Soils. *Soils and Foundations* 42, 1–17.

Sridharan, A., El-Shafei, A., Miura, N., 2002, Mechanisms Controlling the Undrained
Strength Behavior of Remolded Ariake Marine Clays: *Marine Georesources &*
*Geotechnology* 20, 21–50.

Tanaka, M., Tanaka, H., Kamei, T., Hayashi S., 2003. Effects of diatom microfossil
contents on engineering properties of soils, paper presented at Thirteenth
International Offshore and Polar Engineering Conference, Int. Soc. of Offshore
and Polar Eng., Honolulu, Hawaii, 25–30 May.

Tanaka, M., Watabe, Y., Tomita, R., Kamei, T., 2012. Effects of diatom microfossils
content on physical properties of clays. In *ISOPE International Ocean and Polar*
*Engineering Conference* (pp. ISOPE-I). ISOPE.

Urlaub, M., Geersen, J., Krastel, S., Schwenk, T., 2018. Diatom ooze: Crucial for the
generation of submarine mega-slides? *Geology* 46, 331–334.

Wang, R., Clemens, S., Huang, B., Chen, M., 2003. Quaternary palaeoceanographic
changes in the northern South China Sea (ODP Site 1146): radiolarian evidence. *J.*

- Quaternary Sci. 18, 745–756.
- Wiemer, G., Kopf, A., 2017. Influence of diatom microfossils on sediment shear
strength and slope stability. *Geochemistry, Geophysics, Geosystems* 18, 333–345.
- Wroth, C.P., Wood, D.M., 1978, The correlation of index properties with some basic
engineering properties of soils: *Canadian Geotechnical Journal* 15, 137–145.
- Zhang, X., Liu, X., Xu, Y., Wang, G., Ren, Y., 2023. Compressibility, permeability
and microstructure of fine-grained soils containing diatom microfossils.
*Géotechnique*, 74, 661-675.
- Zuluaga, D. A., Sabogal, D., Buenaventura, C. A., Slebi, C.J., 2021. Physical and
mechanical behavior of fine soil according to the content of multispecies diatoms.
*J. Phys. Conf. Ser.* 2118, 012011.
- Zuluaga D. A., Ruge J.C., Camacho-Tauta J., Reyes-Ortiz, O., Caicedo-Hormaza B.,
2022. Diatomaceous soils and advances in geotechnical engineering—Part I.
*Applied Sciences* 13(1), 549.

Reply to Reviewer#2's Comments

Comment 1:

please fix: In Equation 1 (Line 208), the right side should be written as $0.11 + 0.0037PI$. The text has $0.11 + 0.037PI$. I checked the excel workbook in the supplemental material and it is correct there.

Reply 1:

Thanks for pointing this clerical error. We have corrected it in the text.
Please see the Line 199.

Reply to Prof. Davide Gamboa's Comments

Comment 1:

Line 103: very minor comment, but I noticed in figure 2 and also in a figure in the supplementary material that the representation of the well top of ODP Site 1146 is some distance above the seafloor. Maybe it is my bias, but the top/cross in circle often indicates the well datum zero. In ODP/IODP, that is the seafloor, so one could expect for the well top to be coincident with the seafloor reflection – even if for a TWT section. I am aware of how things can be challenging to match synthetic, well and seismic.

If the well path in figure 2B is merely representative of location – you are not clearly showing any well tops not other clear marker, I would suggest moving the top to the seafloor. I am perfectly ok with the way the rest of the figure is presented, and the the synthetics tie in panel A.

Again, this is a minor thing, but for the sake of precise representation, I suggest doing that minor shift on the well representation. The well zero is either seafloor or drill floor (less commonly, sea level), never hanging somewhere on the water column. It may just be a matter of changing the vector edition file. Consider the same for the supplementary material figures

Reply 1:

Thanks for your suggestions. We have made a slight shift, making the well top and well zero correlate to the seafloor in Figure 2 and Figure S9 in the supplementary material.

Comment 2:

Line 108: again on figure 2: as the authors determined the approximate age of the reflection markers, the figure would gain more information if the age estimated for each horizon was included. This could be a simple box with the numbers inside over the corresponding coloured line. Where, up to the authors choice, but all vertically lined up close to the left side of the seismic line could be an option.

Talking about the seismic line, please indicate the orientation SW NE

Reply 2:

As you suggested, we have added the age information for each horizon in Figure 2b.
Also, the orientation has also been added now.

Comment 3:

Add orientations to the seismic profiles of figure 3

Reply 3:

We have added the orientations to the seismic profiles in the Figure 3.

Also, we have added the orientations for all the seismic profiles in the Supplementary Information S1.

Comment 4:

Line 129: I suggest citing figure 1 here as well, as that is the one that shows the distribution of the 16 mtds. Figure 3 is limited to support the statement

Reply 4:

We have cited Figure 1 for supporting the 16 MTDs identification in this study.

Please see the Line 123.

Comment 5:

Line 291: again, another very small thing, but I would suggest adding a box label to the MPT marker/level in figures, or when referring to end/after, remind that it uses SHI as a reference. Until I memorized this, I went a bit back and forth between texts and figures, and this information could be more direct to the reader.

Reply 5:

In Figure 5, we have added a box label showing the MPT interval.

Comment 6:

Line 277: Urlaub et al did not find a statistical correlation... but that does not mean you cannot try to. If all the landslides relate to the weak layers, and these are influenced by climate/sea-level changes, can't that indirectly imply a relationship?

Reply 6:

As you suggested, our results indeed show a correlation between the landslides frequency and climate/sea-level after MPT in the northern SCS margin. This seems to be different from the statistic results from Urlaub et al. (2013), which were mainly focused on the relationship between the landslides and sediment supply variation with sea-level changes, but did not consider too much about other factor variation with sea-level changes. Accordingly, our results indicated that the landslide occurrences could be linked to climate/sea-level changes if other critical factors, such as variations in opal content, are involved.

Now, we have revised this part to highlight that the landslide frequency could be related to climate/sea-level fluctuations when other factor (i.e. opal content) involved.

Please see the details in Line 262-269.

Comment 7:

Line 290: I do not disagree with the sentence, but it seems to make it sound a bit too certain that weak beds do not exist below Late Q. I can guess that the authors may say that this is not necessarily the case, but maybe that needs to be clear. Weak layers likely exist in the sequences

without the mtds, but maybe their composition prevented the high recurrence failures seen in the Late Q, and/or at the scale that can be detected on the seismic (that can open another perspective... speculate on small collapses in MPT that seismic could not pick, except for one case, but post-MPT all conditions favoures bigger ones). Looking at figure 5, a very direct idea would just be the opal content. that can be approached in the discussion as a possible influence for this higher recurrence of mtds in Late Q and not before.

Reply 7:

Based on your suggestion, we have clarify this sentence and expanded the discussion on the formation of opal-enriched weak layers, highlighting the critical role of “opal-enriched layers” in the recurrence of MTDs following the MPT.

Please see the details in Line 272-275, 280-284.

Comment 8:

As a final note, I feel the discussion is falling a bit short on actually linking many of the results together, namely the link with the Geotech results. Maybe a few words on what leads opal to fail, and what favourable set of conditions occurred on this area to lead to such recurrent failures. Even if that is on the results, it would gain if being discussed. All in all, the “core” of discussion addressing the failure aspect is nearly the same length of the conclusions. To which, I may add, are good and refer to the lithological and geotechnical analysis. However, the latter are missing on the discussion.

Reply 8:

As you suggested, we have added the content to explain how the opal-rich layers linked to the weak layers (i.e. the layer with low shear strength), as well as the background of low-magnitude earthquakes. The combined effects of the weak layers and earthquakes resulted in the increased landslide frequency after MPT in the northern SCS margin.

Please see the details in Line 272-284.

Greetings,

There have been substantive efforts on this paper and those efforts are noted. I am supportive of this study in general, but I still have reservations about a key conclusion. In particular, this claim in the abstract “Each horizon marks a rapid increase in opal content, particle size, and porosity, which drives a reduction in the undrained shear strength, thus forming a weak layer,” and the similar claim in the conclusion “These horizons are dominated by elevated opal content, sedimentation rate and thus high porosity, which contributed to the reduction of undrained shear strength and to the formation of weak layers along these horizons.”

As was noted in the previous review and acknowledged by the authors, the actual strength of these horizons was not measured. The authors have now applied a numerical model to simulate the undrained shear strength. However, there are two issues with this technique and the conclusions drawn from it. The higher porosity zone is equated to a lower undrained shear strength. This is not necessarily true for opal/diatom bearing horizons.

The authors are claiming that the presence of the opal and higher grain size are defining a weak layer. It's based on a porosity argument because of the equation used. It is not based on opal content, or even grain size.

It's not explained where the opal is coming from at Site 1146. Is this due primarily diatoms, primarily not diatoms, or some mixture of other biogenic silica (opal) sources? If opal is coming from diatoms, diatoms are known to preserve high porosity while at the same time having higher shear strengths, not lower. This has been detailed by Wiemer and Kopf, 2017 and Shiwakoti et al., 2002. This Wiemer and Kopf paper is not referenced in this manuscript and the Shiwakoti reference is listed but only vaguely saying that opal may influence geotechnical properties (Lines 300-302). Right now, the paper is trying to show weak layers by connecting it to opal, but the literature shows, it's not that simple, and actually suggest it enhances the strength. The rough surface area of the diatom shell (frustule) contributes to high shear strength. At the same time though, the frustule has high intraparticle porosity. Frustules are also in the silt size range. This is also not well-defended to define “coarse” fraction as 19 microns. Typically, the coarse fraction would be defined as 63 microns and above (i.e. sand-sized and coarser). Therefore, it is possible to have high-porosity and high shear strength in diatom-bearing horizons and at the same time being in the “coarse” fraction if one considers 19 microns as coarse.

Because of this, the authors need another mechanism for the weak layers. I'm not sure the authors need to force this connection of enhanced opal equals low shear strength, thus explaining weak layers. It would be better and more fair to the data to state that it's not well understood and offer some suggestions, as opposed to making a firm conclusive statement.

References

Shiwakoti, D.R. , Hiroyuki Tanaka, Masanori Tanaka, J. Locat, Influences of Diatom Microfossils on Engineering Properties of Soils, Soils and Foundations, Volume 42, Issue 3, 2002, Pages 1-17, https://doi.org/10.3208/sandf.42.3_1.

Wiemer, G. and A. Kopf (2017), Influence of diatom microfossils on sediment shear strength and slope stability, *Geochem. Geophys. Geosyst.*, 18, 333–345, doi:[10.1002/2016GC006568](https://doi.org/10.1002/2016GC006568).